# Local online learning in recurrent networks with random feedback

**James M Murray***

Zuckerman Mind, Brain and Behavior Institute, Columbia University, New York, United States

**Abstract** Recurrent neural networks (RNNs) enable the production and processing of time-dependent signals such as those involved in movement or working memory. Classic gradient-based algorithms for training RNNs have been available for decades, but are inconsistent with biological features of the brain, such as causality and locality. We derive an approximation to gradient-based learning that comports with these constraints by requiring synaptic weight updates to depend only on local information about pre- and postsynaptic activities, in addition to a random feedback projection of the RNN output error. In addition to providing mathematical arguments for the effectiveness of the new learning rule, we show through simulations that it can be used to train an RNN to perform a variety of tasks. Finally, to overcome the difficulty of training over very large numbers of timesteps, we propose an augmented circuit architecture that allows the RNN to concatenate short-duration patterns into longer sequences.

DOI: https://doi.org/10.7554/eLife.43299.001

## Introduction

Many tasks require computations that unfold over time. To accomplish tasks involving motor control, working memory, or other time-dependent phenomena, neural circuits must learn to produce the correct output at the correct time. Such learning is a difficult computational problem, as it generally involves temporal credit assignment, requiring synaptic weight updates at a particular time to minimize errors not only at the time of learning but also at earlier and later times. The problem is also a very general one, as such learning occurs in numerous brain areas and is thought to underlie many complex cognitive and motor tasks encountered in experiments.

To obtain insight into how the brain might perform challenging time-dependent computations, an increasingly common approach is to train high-dimensional dynamical systems known as recurrent neural networks (RNNs) to perform tasks similar to those performed by circuits of the brain, often with the goal of comparing the RNN with neural data to obtain insight about how the brain solves computational problems (*Mante et al., 2013*; *Carnevale et al., 2015*; *Sussillo et al., 2015*; *Remington et al., 2018*). While such an approach can lead to useful insights about the neural representations that are formed once a task is learned, it so far cannot address in a satisfying way the process of learning itself, as the standard learning rules for training RNNs suffer from highly nonbiological features such as nonlocality and acausality, as we describe below.

The most straightforward approach to training an RNN to produce a desired output is to define a loss function based on the difference between the RNN output and the target output that we would like it to match, then to update each parameter in the RNN—typically the synaptic weights—by an amount proportional to the gradient of the loss function with respect to that parameter. The most widely used among these algorithms is backpropagation through time (BPTT) (*Rumelhart et al., 1985*). As its name suggests, BPTT is acausal, requiring that errors in the RNN output be accumulated incrementally from the end of a trial to the beginning in order to update synaptic weights. Real-time recurrent learning (RTRL) (*Williams and Zipser, 1989*), the other classic gradient-based

***For correspondence:**
jm4347@columbia.edu

**Competing interests:** The author declares that no competing interests exist.

learning rule, is causal but nonlocal, with the update to a particular synaptic weight in the RNN depending on the full state of the network—a limitation shared by more modern reservoir computing methods (*Jaeger and Haas, 2004*; *Sussillo and Abbott, 2009*). What's more, both BPTT and RTRL require fine tuning in the sense that the feedback weights from the RNN output back to the network must precisely match the readout weights from the RNN to its output. Such precise matching corresponds to fine tuning in the sense that it requires a highly particular initial configuration of the synaptic weights, typically with no justification as to how such a configuration might come about in a biologically plausible manner. Further, if the readout weights are modified during training of the RNN, then the feedback weights must also be updated to match them, and it is unclear how this might be done without requiring nonlocal information.

The goal of this work is to derive a learning rule for RNNs that is both causal and local, without requiring fine tuning of the feedback weights. Our results depend crucially on two approximations. First, locality is enforced by dropping the nonlocal part of the loss function gradient, making our learning rule only approximately gradient-based. Second, we replace the finely tuned feedback weights required by gradient-based learning with random feedback weights, inspired by the success of a similar approach in nonrecurrent feedforward networks (*Lillicrap et al., 2016*; *Liao et al., 2016*). While these two approximations address distinct shortcomings of gradient-based learning and can be made independently (as discussed below in Results), only when both are made together does a learning rule emerge that is fully biologically plausible in the sense of being causal, local, and avoiding fine tuning of feedback weights. In the sections that follow, we show that, even with these approximations, RNNs can be effectively trained to perform a variety of tasks. In the Appendices, we provide supplementary mathematical arguments showing why the algorithm remains effective despite its use of an inexact loss function gradient.

## Results

### The RFLO learning rule

To begin, we consider an RNN, as shown in *Figure 1*, in which a time-dependent input vector $\mathbf{x}(t)$ provides input to a recurrently connected hidden layer of $N$ units described by activity vector $\mathbf{h}(t)$, and this activity is read out to form a time-dependent output $\mathbf{y}(t)$. Such a network is defined by the following equations:

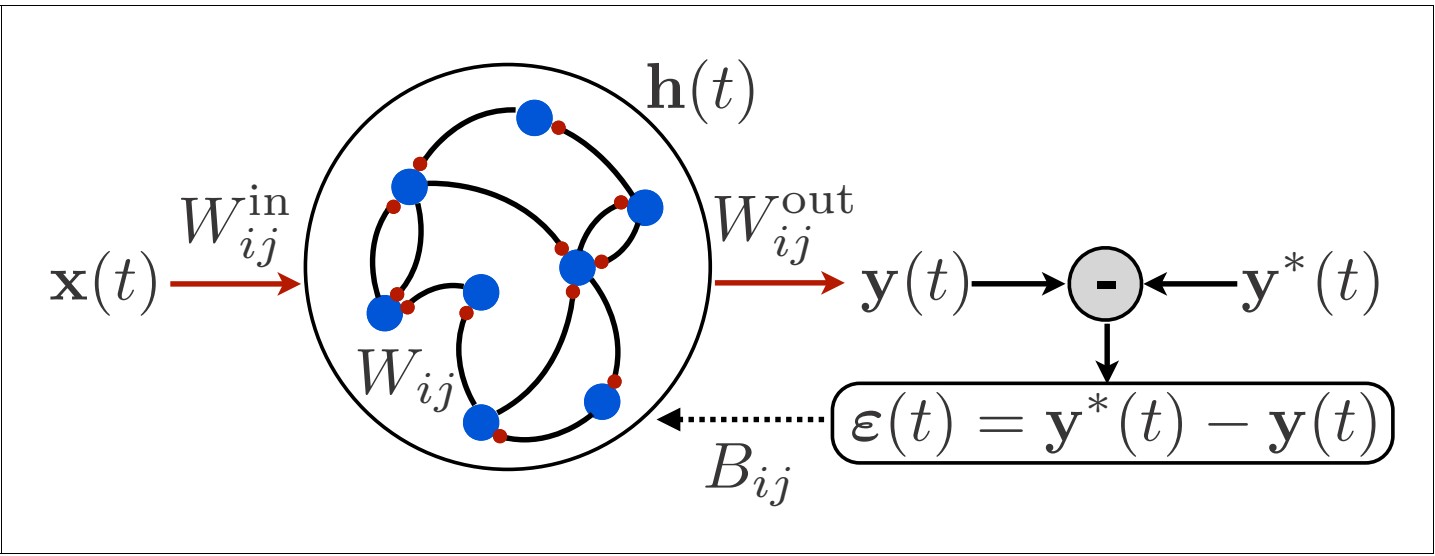

**Figure 1.** Schematic illustration of a recurrent neural network. The network receives time-dependent input $\mathbf{x}(t)$, and its synaptic weights are trained so that the output $\mathbf{y}(t)$ matches a target function $\mathbf{y}^*(t)$. The projection of the error $\varepsilon(t)$ with feedback weights is used for learning the input weights and recurrent weights.
DOI: https://doi.org/10.7554/eLife.43299.002

$$h_i(t+1) = h_i(t) + \frac{1}{\tau}\left[-h_i(t) + \phi\left(\sum_{j=1}^{N} W_{ij}h_j(t) + \sum_{j=1}^{N_x} W_{ij}^{\text{in}}x_j(t+1)\right)\right],$$

$$y_k(t) = \sum_{i=1}^{N} W_{ki}^{\text{out}}h_i(t). \tag{1}$$

For concreteness, we take the nonlinear function appearing in *Equation (1)* to be $\phi(\cdot) = \tanh(\cdot)$. The goal is to train this network to produce a target output function $\mathbf{y}^*(t)$ given a specified input function $\mathbf{x}(t)$ and initial activity vector $\mathbf{h}(0)$. The error is then the difference between the target output and the actual output, and the loss function is the squared error integrated over time:

$$\varepsilon_k(t) = y_k^*(t) - y_k(t),$$

$$L = \frac{1}{2T}\sum_{t=1}^{T}\sum_{k=1}^{N_y}[\varepsilon_k(t)]^2. \tag{2}$$

The goal of producing the target output function $\mathbf{y}^*(t)$ is equivalent to minimizing this loss function.

In order to minimize the loss function with respect to the recurrent weights, we take the derivative with respect to these weights:

$$\frac{\partial L}{\partial W_{ab}} = -\frac{1}{T}\sum_{t=1}^{T}\sum_{j=1}^{N}[(\mathbf{W}^{\text{out}})^{\text{T}}\varepsilon(t)])]_j\frac{\partial h_j(t)}{\partial W_{ab}}. \tag{3}$$

Next, using the update *Equation (1)*, we obtain the following recursion relation:

$$\frac{\partial h_j(t)}{\partial W_{ab}} = \left(1 - \frac{1}{\tau}\right)\frac{\partial h_j(t-1)}{\partial W_{ab}} + \frac{1}{\tau}\delta_{ja}\phi'(u_a(t))h_b(t-1) + \frac{1}{\tau}\sum_k\phi'(u_j(t))W_{jk}\frac{\partial h_k(t-1)}{\partial W_{ab}}, \tag{4}$$

where $\delta_{ja}$ is the Kronecker delta function, $u_a(t)$ is the input current to unit $a$, and the recursion terminates with $\partial h_j(0)/\partial W_{ab} = 0$. This gradient can be updated online at each timestep as the RNN is run, and implementing gradient descent to update the weights using *Equation (3)*, we have $\Delta W_{ab} = -\eta\partial L/\partial W_{ab}$, where $\eta$ is a learning rate. This approach, known as RTRL (*Williams and Zipser, 1989*), is one of the two classic gradient-based algorithms for training RNNs. This approach can also be used for training the input and output weights of the RNN. The full derivation is presented in Appendix 1. (The other classic gradient-based algorithm, BPTT, involves a different approach for taking partial derivatives but is equivalent to RTRL; its derivation and relation to RTRL are also provided in Appendix 1.)

From a biological perspective, there are two problems with RTRL as a plausible rule for synaptic plasticity. The first problem is that it is nonlocal, with the update to synaptic weight $W_{ab}$ depending, through the last term in *Equation (4)*, on every other synaptic weight in the RNN. This information would be inaccessible to a synapse in an actual neural circuit. The second problem is the appearance of $(\mathbf{W}^{\text{out}})^{\text{T}}$ in *Equation (3)*, which means that the error in the RNN output must be fed back into the network with synaptic weights that are precisely symmetric with the readout weights. It is unclear how the readout and feedback weights could be made to match one another in a neural circuit in the brain.

In order to address these two shortcomings, we make two approximations to the RTRL learning rule. The first approximation consists of dropping a nonlocal term from the gradient, so that computing the update to a given synaptic weight requires only pre- and postsynaptic activities, rather than information about the entire state of the RNN including all of its synaptic weights. Second, as described in more detail below, we project the error back into the network for learning using random feedback weights, rather than feedback weights that are tuned to match the readout weights. These approximations, described more fully in Appendix 1, result in the following weight update equations:

$$\delta W_{ab}^{\text{out}}(t) = \eta_1 \varepsilon_a(t) h_b(t),$$
$$\delta W_{ab}(t) = \eta_2 [\mathbf{B}\varepsilon(t)]_a p_{ab}(t),$$
$$\delta W_{ab}^{\text{in}}(t) = \eta_3 [\mathbf{B}\varepsilon(t)]_a q_{ab}(t), \tag{5}$$

where $\eta_\alpha$ are learning rates, and $\mathbf{B}$ is a random matrix of feedback weights. Here we have defined

$$p_{ab}(t) = \frac{1}{\tau}\phi'(u_a(t)) h_b(t-1) + \left(1 - \frac{1}{\tau}\right) p_{ab}(t-1),$$
$$q_{ab}(t) = \frac{1}{\tau}\phi'(u_a(t)) x_b(t-1) + \left(1 - \frac{1}{\tau}\right) q_{ab}(t-1), \tag{6}$$

which are the accumulated products of the pre- and (the derivative of the) postsynaptic activity at the recurrent and input synapses, respectively. We have also defined $u_a(t) \equiv \sum_c W_{ac} h_c(t-1) + \sum_c W_{ac}^{\text{in}} x_c(t)$ as the total input current to unit $a$. While this form of the update equations does not require explicit integration and hence is more efficient for numerical simulation, it is instructive to take the continuous-time ($\tau \gg 1$) limit of *Equation (5)* and the integral of *Equation (6)*, which yields

$$\delta W_{ab}^{\text{out}}(t) = \eta_1 \varepsilon_a(t) h_b(t),$$
$$\delta W_{ab}(t) = \eta_2 [\mathbf{B}\varepsilon(t)]_a \int_0^t \frac{dt'}{\tau} e^{-t'/\tau} \phi'(u_a(t-t')) h_b(t-t'),$$
$$\delta W_{ab}^{\text{in}}(t) = \eta_3 [\mathbf{B}\varepsilon(t)]_a \int_0^t \frac{dt'}{\tau} e^{-t'/\tau} \phi'(u_a(t-t')) x_b(t-t'). \tag{7}$$

In this way, it becomes clear that the integrals in the second and third equations are *eligibility traces* that accumulate the correlations between pre- and post-synaptic activity over a time window of duration $\sim\tau$. The weight update is then proportional to this eligibility trace, multiplied by a feedback projection of the readout error. The fact that the timescale for the eligibility trace matches the RNN time constant $\tau$ reflects the fact that the RNN dynamics are typically correlated only up to this timescale, so that the error is associated only with RNN activity up to time $\tau$ in the past. If the error feedback were delayed rather than provided instantaneously, then eligibility traces with longer timescales might be beneficial (*Gerstner et al., 2018*).

Three features of the above learning rules are especially important. First, the updates are local, requiring information about the presynaptic activity and the postsynaptic input current, but no information about synaptic weights and activity levels elsewhere in the network. Second, the updates are online and can either be made at each timestep or accumulated over many timesteps and made at the end of each trial or of several trials. In either case, unlike the BPTT algorithm, it is not necessary to run the dynamics backward in time at the end of each trial to compute the weight updates. Third, the readout error is projected back to each unit in the network with weights $\mathbf{B}$ that are fixed and random. An exact gradient of the loss function, on the other hand, would lead to $(\mathbf{W}^{\text{out}})^{\text{T}}$, where $(\cdot)^{\text{T}}$ denotes matrix transpose, appearing in the place of $\mathbf{B}$. As described above, the use of random feedback weights is inspired by a similar approach in feedforward networks (*Lillicrap et al., 2016*; see also *Nøkland, 2016*, as well as a recent implementation in feedforward spiking networks [*Samadi et al., 2017*]), and we shall show below that the same feedback alignment mechanism that is responsible for the success of the feedforward version is also at work in our recurrent version. (While an RNN is often described as being 'unrolled in time', so that it becomes a feedforward network in which each layer corresponds to one timestep, it is important to note that the unrolled version of the problem that we consider here is not identical to the feedforward case considered in *Lillicrap et al. (2016)* and *Nøkland, 2016*. In the RNN, a readout error is defined at every 'layer' $t$, whereas in the feedforward case, the error is defined only at the last layer ($t = T$) and is fed back to update weights in all preceding layers.)

With the above observations in mind, we refer to the above learning rule as random feedback local online (RFLO) learning. In Appendix 1, we provide a full derivation of the learning rule, and describe in detail its relation to the other gradient-based methods mentioned above, BPTT and RTRL. It should be noted that the approximations applied above to the RTRL algorithm are distinct from recent approximations made in the machine learning literature (*Tallec and Ollivier, 2018*;

*Mujika et al., 2018*), where the goal was to decrease the computational cost of RTRL, rather than to increase its biological plausibility.

Because the RFLO learning rule uses an approximation of the loss function gradient rather than the exact gradient for updating the synaptic weights, a natural question to ask is whether it can be expected to decrease the loss function at all. In Appendix 2 we show that, under certain simplifying assumptions including linearization of the RNN, the loss function does indeed decrease on average with each step of RFLO learning. In particular, we show that, as in the feedforward case (*Lillicrap et al., 2016*), reduction of the loss function requires alignment between the learned readout weights $\mathbf{W}^{\text{out}}$ and the fixed feedback weights $\mathbf{B}$. We then proceed to show that this alignment tends to increase during training due to coordinated learning of the recurrent weights $\mathbf{W}$ and readout weights $\mathbf{W}^{\text{out}}$. The mathematical approach for showing that alignment between readout and feedback weights occurs is similar to that used previously in the feedforward case (*Lillicrap et al., 2016*). In particular, the network was made fully linear in both cases in order to make mathematical headway possible, and a statistical average over inputs (in the feedforward case) or the activity vector (for the RNN) was performed. However, because a feedforward network retains no state information from one timestep to the next and because the network architectures are distinct (even if one thinks about an RNN as a feedforward network 'unrolled in time'), the results in Appendix 2 are not simply a straightforward generalization of the feedforward case.

A number of simplifying assumptions have been made in the mathematical derivations of Appendix 2, including linear dynamics, uncorrelated neurons, and random synaptic weights, none of which will necessarily hold in a nonlinear network trained to perform a dynamical computation. Hence, although such mathematical arguments provide reason to hope that RFLO learning might be successful and insight into the mechanism by which learning occurs, it remains to be shown that RFLO learning can be used to successfully train a nonlinear RNN in practice. In the following section, therefore, we show using simulated examples that RFLO learning can perform well on a variety of tasks.

## Performance of RFLO learning

In this section we illustrate the performance of the RFLO learning algorithm on a number of simulated tasks. These tasks require an RNN to produce sequences of output values and/or delayed responses to an input to the RNN, and hence are beyond the capabilities of feedforward networks. As a benchmark, we compare the performance of RFLO learning with BPTT, the standard algorithm for training RNNs. (As described in Appendix 1, the weight updates in RTRL are, when performed in batches at the end of each trial, completely equivalent to those in BPTT. Hence in this section we compare RFLO learning with BPTT only in what follows.)

### Autonomous production of continuous outputs

*Figure 2* illustrates the performance of an RNN trained with RFLO learning to produce a one-dimensional periodic output given no external input. *Figure 2a* shows the decrease of the loss function (the mean squared error of the RNN output) as the RNN is trained over many trials, where each trial corresponds to one period consisting of $T$ timesteps, as well as the performance of the RNN at the end of training. As a benchmark for comparison with the RFLO learning rule, BPTT was also used to train the RNN. In addition, we show in *Figure 2—figure supplement 1* that a variant of RFLO learning in which all outbound synapses from a given unit were constrained to be of the same sign—a biological constraint known as Dale's law (*Dale, 1935*)—also yields effective learning. (A similar result, in this case using nonlocal learning rules, was recently obtained in other modeling work [*Song et al., 2016*].)

*Figure 2b* shows that, in the case where the number of timesteps in the target output was not too great, both versions of RFLO learning perform comparably well to BPTT. BPTT shows an advantage, however, when the number of timesteps became very large. Intuitively, this difference in performance is due to the accumulation of small errors in the estimated gradient of the loss function over many timesteps with RFLO learning. This is less of a problem for BPTT, on the other hand, in which the exact gradient is used.

*Figure 2c* shows the increase in the alignment between the vector of readout weights $\mathbf{W}^{\text{out}}$ and the vector of feedback weights $\mathbf{B}$ during training with RFLO learning. As in the case of feedforward networks (*Lillicrap et al., 2016*; *Nøkland, 2016*), the readout weights evolve over time to become

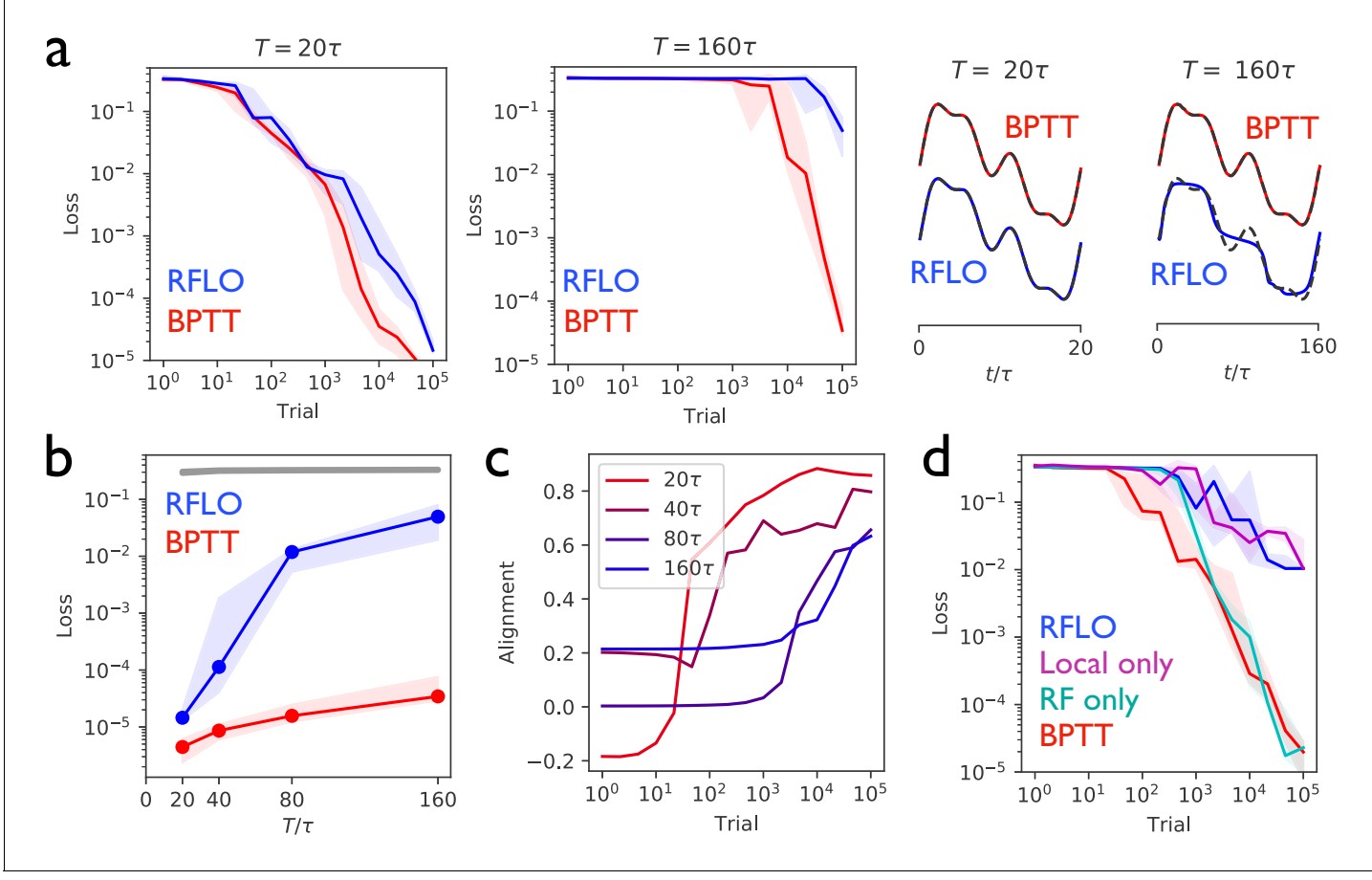

**Figure 2.** Periodic output task. (**a**) *Left panels:* The mean squared output error during training for an RNN with $N = 30$ recurrent units and no external input, trained to produce a one-dimensional periodic output with period of duration $T = 20\tau$ (left) or $T = 160\tau$ (right), where $\tau = 10$ is the RNN time constant. The learning rules used for training were backpropagation through time (BPTT) and random feedback local online (RFLO) learning. Solid line is median loss over nine realizations, and shaded regions show 25/75 percentiles. *Right panels:* The RNN output at the end of training for each type of learning (dashed lines are target outputs, offset for clarity). (**b**) The loss function at the end of training for target outputs having different periods. The colored lines correspond to the two learning rules from (**a**), while the gray line is the loss computed for an untrained RNN. (**c**) The normalized alignment between the vector of readout weights $\mathbf{W}^{\text{out}}$ and the vector of feedback weights $\mathbf{B}$ during training with RFLO learning. (**d**) The loss function during training with $T = 80\tau$ for BPTT and RFLO, as well as versions of RFLO in which locality is enforced without random feedback (magenta) or random feedback is used without enforcing locality (cyan).

DOI: https://doi.org/10.7554/eLife.43299.003

The following figure supplements are available for figure 2:

**Figure supplement 1.** An RNN with sign-constrained synapses comporting with Dale's law attains performance similar to an unconstrained RNN.

DOI: https://doi.org/10.7554/eLife.43299.004

**Figure supplement 2.** An RNN trained to perform the task from *Figure 2* with RFLO learning on recurrent and readout weights outperforms an RNN in which only readout weights or only recurrent weights are trained.

DOI: https://doi.org/10.7554/eLife.43299.005

**Figure supplement 3.** The performance of an RNN trained to perform the task from *Figure 2* with RFLO learning improves with larger network sizes and larger initial recurrent weights.

DOI: https://doi.org/10.7554/eLife.43299.006

increasingly similar to the feedback weights, which are fixed during training. In Appendix 2 we provide mathematical arguments for why this alignment occurs, showing that the alignment is not due to the change in $\mathbf{W}^{\text{out}}$ alone, but rather to coordinated changes in the readout and recurrent weights.

In deriving the RFLO learning rule, two independent approximations were made: locality was enforced by dropping the nonlocal term from the loss function gradient, and feedback weights were

chosen randomly rather than tuned to match the readout weights. If these approximations are instead made independently, which will have the greater effect on the performance of the RNN? *Figure 2d* answers this question by comparing RFLO and BPTT with two alternative learning rules: one in which the local approximation is made while symmetric error feedback is maintained, and another in which the nonlocal part of the loss function gradient is retained but the error feedback is random. The results show that the local approximation is essentially fully responsible for the performance difference between RFLO and BPTT, while there is no significant loss in performance due to the random feedback alone.

It is also worthwhile to consider the relative contributions of the two types of learning in *Figure 2*, namely the learning of recurrent and of readout weights. Given that the learning rule for the readout weights makes use of the exact loss function gradient while that for the recurrent weights does not, it could be that the former are fully responsible for the successful training. In *Figure 2—figure supplement 2* we show that this is not the case, and that training of both recurrent and readout weights significantly outperforms training of the readout weights only (with the readout fed back as an input to the RNN for stability–see Materials and methods). Also shown is the performance of an RNN in which recurrent weights but not readout weights are trained. In this case learning is completely unsuccessful. The reason is that, in order for successful credit assignment to take place, there must be some alignment between the readout weights and feedback weights. Such alignment can't occur, however, if the readout weights are frozen. In the case of a linearized network, the necessity of coordinated learning between the two sets of weights can be shown mathematically, as done in Appendix 2.

As with other RNN training methods, performance of the trained RNN generally improves for larger network sizes (*Figure 2—figure supplement 3*). While the computational cost of training the RNN increases with RNN size, leading to a tradeoff between fast training and high performance for a given number of training trials, it is worthwhile to note that the cost is much lower than that of RTRL ($\sim N^4$ operations per timestep) and is on par with BPTT (both $\sim N^2$ operations per timestep, as shown in Appendix 1).

## Interval matching

*Figure 3* illustrates the performance of the RFLO algorithm on a 'Ready Set Go' task, in which the RNN is required to produce an output pulse after a time delay matching the delay between two preceding input pulses (*Jazayeri and Shadlen, 2010*). This task is more difficult than the production of

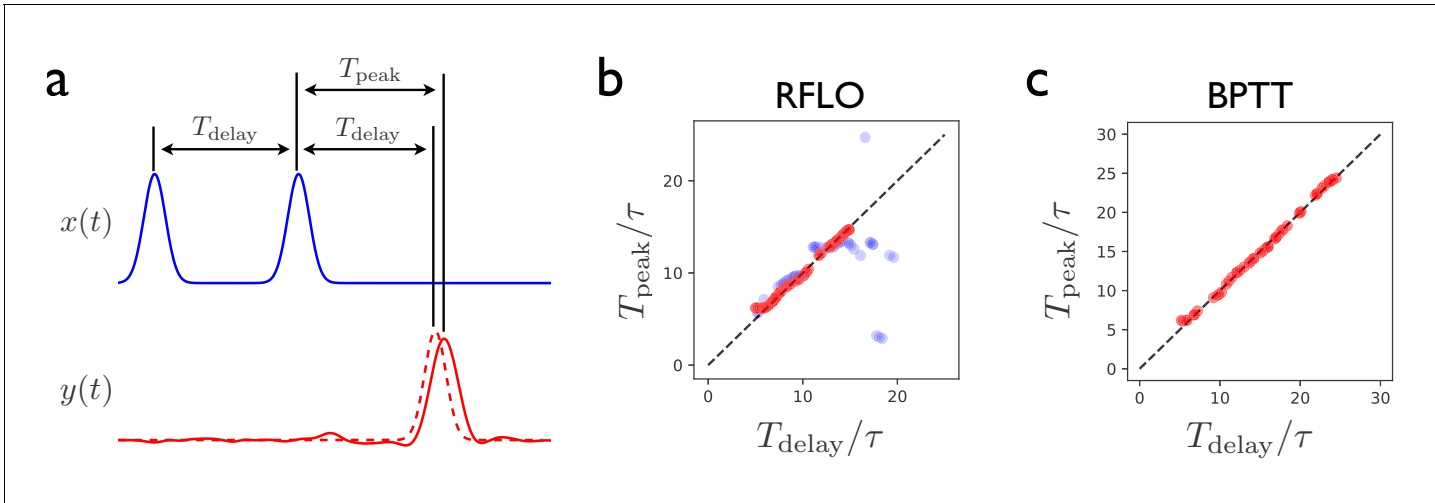

**Figure 3.** Interval-matching task. (a) In the task, the RNN input consists of two input pulses, with a random delay $T_{\text{delay}}$ between pulses in each trial. The target output (dashed line) is a pulse trailing the second input pulse by $T_{\text{delay}}$. (b) The time of the peak in the RNN output is observed after training with RFLO learning and testing in trials with various interpulse delays in the input. Red (blue) shows the case in which the RNN is trained with interpulse delays satisfying $T_{\text{delay}} \leq 15\tau$ ($20\tau$). (c) Same as (b), but with the RNN trained using BPTT using interpulse delays $T_{\text{delay}} \leq 25\tau$ for training and testing.
DOI: https://doi.org/10.7554/eLife.43299.007

a periodic output due to the requirement that the RNN must learn to store the information about the interpulse delay, and then produce responses at different times depending on what the delay was. *Figure 3b,c* illustrate the testing performance of an RNN trained with either RFLO learning or BPTT. If the RNN is trained and tested on interpulse delays satisfying $T_{\text{delay}} \leq 15\tau$, the performance is similarly good for the two algorithms. If the RNN is trained and tested with longer $T_{\text{delay}}$, however, then BPTT performs better than RFLO learning. As in the case of the periodic output task from *Figure 2*, RFLO learning performs well for tasks on short and intermediate timescales, but not as well as BPTT for tasks involving longer timescales. In the following subsection, we shall address this shortcoming by constructing a network in which learned subsequence elements of short duration can be concatenated to form longer-duration sequences.

## Learning a sequence of actions

In the above examples, it was shown that, while the performance of RFLO learning is comparable to that of BPTT for tasks over short and intermediate timescales, it is less impressive for tasks involving longer timescales. From the perspective of machine learning, this represents a failure of RFLO learning. From the perspective of neuroscience, however, we can adopt a more constructive attitude. The brain, after all, suffers the same limitations that we have imposed in constructing the RFLO learning rule—namely, causality and locality—and cannot be performing BPTT for learned movements and working memory tasks over long timescales of seconds or more. So how might recurrent circuits in the brain learn to perform tasks over these long timescales? One possibility is that they use a more sophisticated learning rule than the one that we have constructed. While we cannot rule out this possibility, it is worth keeping in mind that, due to the problem of vanishing or exploding gradients, all gradient-based training methods for RNNs fail eventually at long timescales. Another possibility is that a simple, fully connected recurrent circuit in the brain, like an RNN trained with RFLO learning, can only be trained directly with supervised learning over short timescales, and that a more complex circuit architecture is necessary for longer timescales.

It has long been recognized that long-duration behaviors tend to be sequences composed of short, stereotyped actions concatenated together (*Lashley, 1951*). Further, a great deal of experimental work suggests that learning of this type involves training of synaptic weights from cortex to striatum (*Graybiel, 1998*), the input structure of the basal ganglia, which in turn modifies cortical activity via thalamus. In this section we propose a circuit architecture, largely borrowed from *Logiaco et al. (2018)* and inspired by the subcortical loop involving basal ganglia and thalamus, that allows an RNN to learn and perform sequences of 'behavioral syllables'.

As illustrated in *Figure 4a*, the first stage of learning in this scheme involves training an RNN to produce a distinct time-dependent output in response to the activation of each of its tonic inputs. In this case, the RNN output is a two-dimensional vector giving the velocity of a cursor moving in a plane. Once the RNN has been trained in this way, the circuit is augmented with a loop structure, shown schematically in *Figure 4b*. At one end of the loop, the RNN activity is read out with weights $\mathbf{W}^s$. At the other end of the loop, this readout is used to control the input to the RNN. The weights $\mathbf{W}^s$ can be learned such that, at the end of one behavioral syllable, the RNN input driving the next syllable in the sequence is activated by the auxiliary loop. This is done most easily by gating the RNN readout so that it can only drive changes at the end of a syllable.

In this example, each time the end of a syllable is reached, four readout units receive input $z_i = \sum_{j=1}^{N} W_{ij}^s h_j$, and a winner-take-all rule is applied such that the most active unit activates a corresponding RNN input unit, which drives the RNN to produce the next syllable. Meanwhile, the weights are updated with the reward-modulated Hebbian learning rule $\Delta W_{ij}^s = \eta_s R z_i h_j$, where $R = 1$ if the syllable transition matches the target and $R = 0$ otherwise. By training over many trials, the network learns to match the target sequence of syllables. *Figure 4c* shows the output from an RNN trained in this way to produce a sequence of reaches and holds in a two-dimensional space. Importantly, while the duration of each behavioral syllable in this example ($20\tau$) is relatively short, the full concatenated sequence is long ($160\tau$) and would be very difficult to train directly in an RNN lacking such a loop structure.

How might the loop architecture illustrated in *Figure 4* be instantiated in the brain? For learned motor control, motor cortex likely plays the role of the recurrent circuit controlling movements. In addition to projections to spinal cord for controlling movement directly, motor cortex also projects

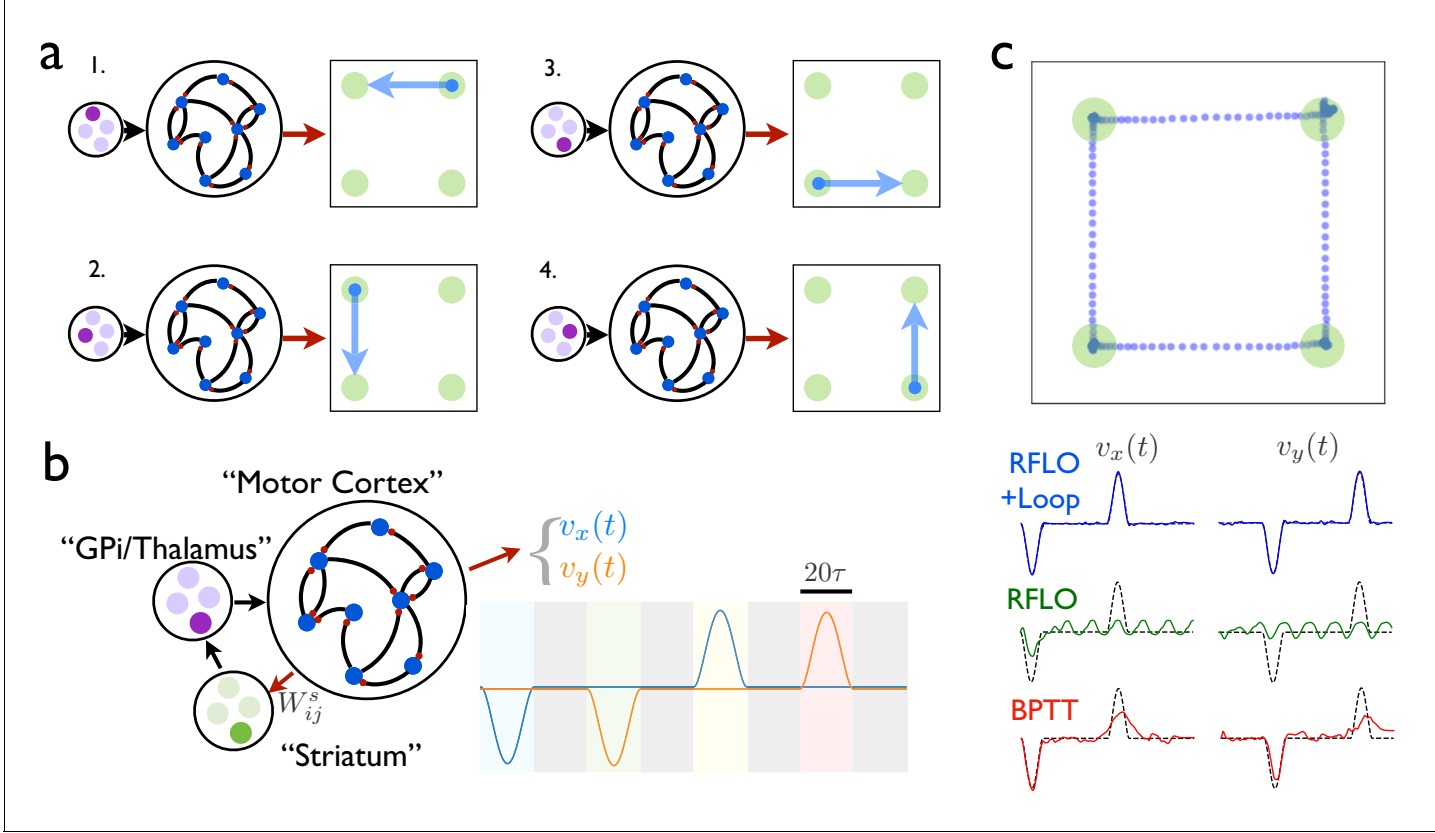

**Figure 4.** An RNN with multiple inputs controlled by an auxiliary loop learns to produce sequences. (**a**) An RNN with a two-dimensional readout controlling the velocity of a cursor is trained to move the cursor in a different direction for each of the four possible inputs. (**b**) The RNN is augmented with a loop structure, which allows a readout from the RNN via learned weights $\mathbf{W}^s$ to change the state of the input to the RNN, enabling the RNN state at the end of each cursor movement to trigger the beginning of the next movement. (**c**) The trajectory of a cursor performing four movements and four holds, where RFLO learning was used to train the individual movements as in (**a**), and learning of the weights $\mathbf{W}^s$ was used to join these movements into a sequence, as illustrated in (**b**). Lower traces show comparison of this trajectory with those obtained by using either RFLO or BPTT to train an RNN to perform the entire sequence without the auxiliary loop.

DOI: https://doi.org/10.7554/eLife.43299.008

to striatum, and experimental evidence has suggested that modification of these corticostriatal synapses plays an important role in the learning of action sequences (*Jin and Costa, 2010*). Via a loop through the basal ganglia output nucleus GPi and motor thalamus, these signals pass back to motor cortex, as illustrated schematically in *Figure 4*. According to the model, then, behavioral syllables are stored in motor cortex, and the role of striatum is to direct the switching from one syllable to the next. Experimental evidence for both the existence of behavioral syllables and the role played by striatum in switching between syllables on subsecond timescales has been found recently in mice (*Wiltschko et al., 2015*; *Markowitz et al., 2018*). How might the weights from motor cortex in this model be gated so that this projection is active at behavioral transitions? It is well known that dopamine, in addition to modulating plasticity at corticostriatal synapses, also modulates the gain of cortical inputs to striatum (*Gerfen et al., 2011*). Further, it has recently been shown that transient dopamine signals occur at the beginning of each movement in a lever-press sequence in mice (*da Silva et al., 2018*). Together, these experimental results support a model in which dopamine bursts enable striatum to direct switching between behavioral syllables, thereby allowing for learned behavioral sequences to occur over long timescales by enabling the RNN to control its own input. Within this framework, RFLO learning provides a biologically plausible means by which the behavioral syllables making up these sequences might be learned.

## Discussion

In this work we have derived an approximation to gradient-based learning rules for RNNs, yielding a learning rule that is local, online, and does not require fine tuning of feedback weights. We have shown that RFLO learning performs comparably well to BPTT when the duration of the task being trained is not too long, but that it performs less well when the task duration becomes very long. In this case, however, we showed that training can still be effective if the RNN architecture is augmented to enable the concatenation of short-duration outputs into longer output sequences. Further exploring how this augmented architecture might map onto cortical and subcortical circuits in the brain is an interesting direction for future work. Another promising area for future work is the use of layered recurrent architectures, which occur throughout cortex and have been shown to be beneficial in complex machine learning applications spanning long timescales (*Pascanu et al., 2014*). Finally, machine learning tasks with discrete timesteps and discrete outputs such as text prediction benefit greatly from the use of RNNs with cross-entropy loss functions and softmax output normalization. In general, these lead to additional nonlocal terms in gradient-based learning, and in future work it would be interesting to investigate whether RFLO learning can be adapted and applied to such problems while preserving locality, or whether new ideas are necessary about how such tasks are solved in the brain.

How might RFLO learning be implemented concretely in the brain? As we have discussed above, motor cortex is an example of a recurrent circuit that can be trained to produce a particular time-dependent output. Neurons in motor cortex receive information about planned actions ($\mathbf{y}^*(t)$ in the language of the model) from premotor cortical areas, as well as information about the current state of the body ($\mathbf{y}(t)$) from visual and/or proprioceptive inputs, giving them the information necessary to compute a time-dependent error $\varepsilon(t) = \mathbf{y}^*(t) - \mathbf{y}(t)$. Hence it is possible that neurons within motor cortex might use a projection of this error signal to learn to produce a target output trajectory. Such a computation might feature a special role for apical dendrites, as in recently developed theories for learning in feedforward cortical networks (*Guerguiev et al., 2017*; *Sacramento et al., 2017*), though further work would be needed to build a detailed theory for its implementation in recurrent cortical circuits.

A possible alternative scenario is that neuromodulators might encode error signals. In particular, midbrain dopamine neurons project to many frontal cortical areas including prefrontal cortex and motor cortex, and their input is known to be necessary for learning certain time-dependent behaviors (*Hosp et al., 2011*; *Li et al., 2017*). Further, recent experiments have shown that the signals encoded by dopamine neurons are significantly richer than the reward prediction error that has traditionally been associated with dopamine, and include phasic modulation during movements (*Howe and Dombeck, 2016*; *da Silva et al., 2018*; *Coddington and Dudman, 2018*). This interpretation of dopamine as a continuous online error signal used for supervised learning would be distinct from and complementary to its well known role as an encoder of reward prediction error for reinforcement learning.

In addition to the gradient-based approaches (RTRL and BPTT) already discussed above, another widely used algorithm for training RNNs is FORCE learning (*Sussillo and Abbott, 2009*) and its more recent variants (*Laje and Buonomano, 2013*; *DePasquale et al., 2018*). The FORCE algorithm, unlike gradient-based approaches, makes use of chaotic fluctuations in RNN activity driven by strong recurrent input. These chaotic fluctuations, which are not necessary in gradient-based approaches, provide a temporally rich set of basis functions that can be summed together with trained readout weights in order to construct a desired time-dependent output. As with gradient-based approaches, however, FORCE learning is nonlocal, in this case because the update to any given readout weight depends not just on the presynaptic activity, but also on the activities of all other units in the network. Although FORCE learning is biologically implausible due to the nonlocality of the learning rule, it is, like RFLO learning, implemented online and does not require finely tuned feedback weights for the readout error. It is an open question whether approximations to the FORCE algorithm might exist that would obviate the need for nonlocal learning while maintaining sufficiently good performance.

In addition to RFLO learning, a number of other local and causal learning rules for training RNNs have been proposed. The oldest of these algorithms (*Mazzoni et al., 1991*; *Williams, 1992*) operate within the framework of reinforcement learning rather than supervised learning, meaning that only a

scalar—and possibly temporally delayed—reward signal is available for training the RNN, rather than the full target function $y^*(t)$. Typical of such algorithms, which are often known as 'node perturbation' algorithms, is the REINFORCE learning rule (*Williams, 1992*), which in our notation gives the following weight update at the end of each trial:

$$\Delta W_{ab} = \frac{\eta}{T}(R - \bar{R}) \sum_{t=1}^{T} \xi_a(t) h_b(t), \tag{8}$$

where $R$ is the scalar reward signal (which might be defined as the negative of the loss function that we have used in RFLO learning), $\bar{R}$ is the average reward over recent trials, and $\xi_a(t)$ is noise current injected into unit $a$ during training. This learning rule means, for example, that (assuming the presynaptic unit $b$ is active) if the postsynaptic unit $a$ is more active than usual in a given trial (i.e. $\xi_a(t)$ is positive) and the reward is greater than expected, then the synaptic weight $W_{ab}$ should be increased so that this postsynaptic unit should be more active in future trials. A slightly more elaborate version of this learning rule replaces the summand in *Equation (8)* with a low-pass filtered version of this same quantity, leading to eligibility traces of similar form to those appearing in *Equation (7)*. This learning rule has also been adapted for a network of spiking neurons (*Fiete et al., 2006*).

A potential shortcoming of the REINFORCE learning rule is that it depends on the postsynaptic noise current rather than on the total postsynaptic input current (i.e. the noise current plus the input current from presynaptic units). Because it is arguably implausible that a neuron could keep track of these sources of input current separately, a recently proposed version (*Miconi, 2017*) replaces $\xi_a(t) \rightarrow f(u_a(t) - \bar{u}_a(t))$, where $f(\cdot)$ is a supralinear function, $u_a(t)$ is the total input current (including noise) to unit $a$, and $\bar{u}_a(t)$ is the low-pass-filtered input current. This substitution is logical since the quantity $u_a(t) - \bar{u}_a(t)$ tracks the fast fluctuations of each unit, which are mainly due to the rapidly fluctuating input noise rather than to the more slowly varying recurrent and feedforward inputs.

A severe limitation of reinforcement learning as formulated in *Equation (8)* is the sparsity of reward information, which comes in the form of a single scalar value at the end of each trial. Clearly this provides the RNN with much less information to learn from than a vector of errors $\varepsilon(t) \equiv \mathbf{y}^*(t) - \mathbf{y}(t)$ at every timestep, which is assumed to be available in supervised learning. As one would expect from this observation, reinforcement learning is typically much slower than supervised learning in RNNs, as in feedforward neural networks. A hybrid approach is to assume that reward information is scalar, as in reinforcement learning, but available at every timestep, as in supervised learning. This might correspond to setting $R(t) \equiv -|\varepsilon(t)|^2$ and including this reward in a learning rule such as the REINFORCE rule in *Equation (8)*. To our knowledge this has not been done for training recurrent weights in an RNN, though a similar idea has recently been used for training the readout weights of an RNN (*Legenstein et al., 2010*; *Hoerzer et al., 2014*). Ultimately, whether recurrent neural circuits in the brain use reinforcement learning or supervised learning is likely to depend on the task being learned and what feedback information about performance is available. For example, in a reach-to-target task such as the one modeled in *Figure 4*, it is plausible that a human or nonhuman primate might have a mental template of an ideal reach, and might make corrections to make the hand match the target trajectory at each timepoint in the trial. On the other hand, if only delayed binary feedback is provided in an interval-matching task such as the one modeled in *Figure 3*, neural circuits in the brain might be more likely to use reinforcement learning.

More recently, local, online algorithms for supervised learning in RNNs with spiking neurons have been proposed. *Gilra and Gerstner (2017)* and *Alemi et al. (2017)* have trained spiking RNNs to produce particular dynamical trajectories of RNN readouts. These works constitute a large step toward greater biological plausibility, particularly in their use of local learning rules and spiking neurons. Here we describe the most important differences between those works and RFLO learning. In both *Gilra and Gerstner (2017)* and *Alemi et al. (2017)*, the RNN is driven by an input $\mathbf{x}(t)$ as well as the error signal $\varepsilon(t) = \mathbf{y}^*(t) - \mathbf{y}(t)$, where the target output is related to the input $\mathbf{x}(t)$ according to

$$\dot{y}_i^* = f_i(\mathbf{y}^*) + g_i(\mathbf{x}), \tag{9}$$

where $g_i(\mathbf{x}) = x_i(t)$ in *Alemi et al. (2017)*, but is arbitrary in *Gilra and Gerstner (2017)*. In either case, however, it is not possible to learn arbitrary, time-dependent mappings between inputs and

outputs in these networks, since the RNN output must take the form of a dynamical system driven by the RNN input. This is especially limiting if one desires that the RNN dynamics should be autonomous, so that $\mathbf{x}(t) = 0$ in *Equation (9)*. It is not obvious, for example, what dynamical equations having the form of (9) would provide a solution to the interval-matching task studied in *Figure 3*. Of course, it is always possible to obtain an arbitrarily complex readout by making $\mathbf{x}(t)$ sufficiently large such that $\mathbf{y}(t)$ simply follows $\mathbf{x}(t)$ from *Equation (9)*. However, since $\mathbf{x}(t)$ is provided as input, the RNN essentially becomes an autoencoder in this limit.

Two other features of *Gilra and Gerstner (2017)* and *Alemi et al. (2017)* differ from RFLO learning. First, the readout weights and the error feedback weights are related to one another in a highly specific way, being either symmetric with one another (*Alemi et al., 2017*), or else configured such that the loop from the RNN to the readout and back to the RNN via the error feedback pathway forms an autoencoder (*Gilra and Gerstner, 2017*). In either case these weights are preset to these values before training of the RNN begins, unlike the randomly set feedback weights used in RFLO learning. Second, both approaches require that the error signal $\varepsilon(t)$ be fed back to the network with (at least initially) sufficiently large gain such that the RNN dynamics are essentially slaved to produce the target readout $\mathbf{y}^*(t)$, so that one has $\mathbf{y}(t) \approx \mathbf{y}^*(t)$ immediately from the beginning of training. (This follows as a consequence of the relation between the readout and feedback weights described above.) With RFLO learning, in contrast, forcing the output to always follow the target in this way is not necessary, and learning can work even if the RNN dynamics early in learning do not resemble the dynamics of the ultimate solution.

In summary, the random feedback learning rule that we propose offers a potential advantage over previous biologically plausible learning rules by making use of the full time-dependent, possibly multidimensional error signal, and also by training all weights in the network, including input, output, and recurrent weights. In addition, it does not require any special relation between the RNN inputs and outputs, nor any special relationship between the readout and feedback weights, nor a mechanism that restricts the RNN dynamics to always match the target from the start of training. Especially when extended to allow for sequence learning such as depicted in *Figure 4*, RFLO learning provides a plausible mechanism by which supervised learning might be implemented in recurrent circuits in the brain.

## Materials and methods

### Source code

A Python notebook implementing a simple, self-contained example of RFLO learning has been included as *Source code 1* to accompany this publication. The example trains an RNN on the periodic output task from *Figure 2* using RFLO learning, as well as using BPTT and RTRL for comparison.

### Simulation details

In all simulations, the RNN time constant was $\tau = 10$. Learning rates were selected by grid search over $\eta_{1,2,3} = \eta \in [10^{-4}, 3 \times 10^{-4}, 10^{-3}, \ldots, 3 \times 10^{-1}]$. Input and readout weights were initialized randomly and uniformly over $[-1, 1]$ and $[-1/\sqrt{N}, 1/\sqrt{N}]$, respectively. Recurrent weights were initialized randomly as $W \sim \mathcal{N}(0, g^2/N)$, where $g = 1.5$ and $\mathcal{N}(0, \sigma^2)$ is the normal distribution with zero mean and variance $\sigma^2$. The fixed feedback weights were chosen randomly as $B_{ij} \sim \mathcal{N}(0, 1)$. The nonlinear activation function of the RNN units was $\phi(\cdot) = \tanh(\cdot)$.

In *Figure 2*, the RNN size was $N = 30$. For task durations of $T = (200, 400, 800, 1600)$ timesteps, the optimal learning rates after grid search were $\eta = (0.03, 0.01, 0.001, 0.0003)$ for RFLO and $(0.03, 0.03, 0.01, 0.03)$ for BPTT. The target output waveform was $y^*(t) = \sin(2\pi t/T) + 0.5\sin(4\pi t/T) + 0.25\sin(8\pi t/T)$. The shaded regions in panels a, b, and d are 25/75 percentiles of performance computed over nine randomly initialized networks, and the solid curves show the median performance.

In the version of the periodic output task satisfying Dale's law enforcing sign-constrained synapses (*Figure 2—figure supplement 1*), half of RNN units were assigned to be excitatory and half were inhibitory. Recurrent weights were initialized as above, with the additional step of $W_{ij} \leftarrow \xi_j|W_{ij}|$,

where $\xi_j = \pm 1$ for excitatory or inhibitory units. During learning in this network, recurrent weights were updated normally but clipped to zero to prevent the weights from changing sign.

In the version of the periodic output task in which only readout weights were trained (*Figure 2—figure supplement 2*), the readout was fed back into the RNN as a separate input current to the recurrent units via the random feedback weights $\mathbf{B}$. This is necessary to stabilize the RNN dynamics in the absence of learning of the recurrent weights, as they would be either chaotic (for large recurrent weights) or quickly decaying (for small recurrent weights) in the absence of such stabilization. The RNN was initialized as described above, and the learning rate for the readout weights was $\eta = 0.03$, determined by grid search.

In *Figure 3*, the RNN size was $N = 100$. The input and target output pulses were Gaussian with a standard deviation of 15 timesteps. The RNNs were trained for 5000 trials. With BPTT, the learning rate was $\eta_{1,2,3} = 0.003$, while with RFLO learning it was $0.001$. Rather than performing weight updates in every trial, the updates were continuously accumulated but only implemented after batches of 10 trials.

In *Figure 4*, networks of size $N = 100$ were used. In the version with the loop architecture, RFLO learning was first used to train the network to produce a particular reach trajectory in response to each of four tonic inputs for 10,000 trials, with a random input chosen in each trial, subject to the constraint that the trajectory could not move the cursor out of bounds. Next, the RNN weights were held fixed and the weights $\mathbf{W}^s$ were learned for 10,000 additional trials while the RNN controlled its own input via the auxiliary loop. The active unit in 'striatum' was chosen randomly with probability $p_{\text{explore}} = 0.1$ and was otherwise chosen deterministically based on the RNN input via the weights $W^s$, again subject to the constraint that the trajectory could not move the cursor out of bounds. In the comparison shown in subpanel (c), RNNs without the loop architecture were trained for 20,000 trials with either RFLO learning or BPTT to autonomously produce the entire sequence of $160\tau$ timesteps.

## Acknowledgements

The author is grateful to LF Abbott, GS Escola, and A Litwin-Kumar for helpful discussions and feedback on the manuscript. Support for this work was provided by the National Science Foundation NeuroNex program (DBI-1707398), the National Institutes of Health (DP5 OD019897), and the Gatsby Charitable Foundation.

## Additional information

### Funding

| Funder | Grant reference number | Author |
| --- | --- | --- |
| National Institutes of Health | DP5 OD019897 | James M Murray |
| National Science Foundation | DBI-1707398 | James M Murray |
| Gatsby Charitable Foundation | | James M Murray |

The funders had no role in study design, data collection and interpretation, or the decision to submit the work for publication.

### Author contributions

James M Murray, Conceptualization, Writing—original draft, Writing—review and editing

### Author ORCIDs

James M Murray ⓘ https://orcid.org/0000-0003-3706-4895

### Decision letter and Author response

Decision letter https://doi.org/10.7554/eLife.43299.014
Author response https://doi.org/10.7554/eLife.43299.015

## Additional files

### Supplementary files

• Source code 1. Example code implementing RFLO learning and BPTT.
DOI: https://doi.org/10.7554/eLife.43299.009

• Transparent reporting form
DOI: https://doi.org/10.7554/eLife.43299.010

### Data availability

Code implementing the RFLO learning algorithm for the example shown in Figure 2 has been included as a source code file accompanying this manuscript.

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

## Appendix 1

DOI: https://doi.org/10.7554/eLife.43299.011

## Gradient-based RNN learning and RFLO learning

In the first subsection of this appendix, we begin by reviewing the derivation of RTRL, the classic gradient-based learning rule. We show that the update equation for the recurrent weights under the RTRL rule has two undesirable features from a biological point of view. First, the learning rule is nonlocal, with the update to weight $W_{ij}$ depending on all of the other weights in the RNN, rather than just on information that is locally available to that particular synapse. Second, the RTRL learning rule requires that the error in the RNN readout be fed back into the RNN with weights that are precisely symmetric with the readout weights. In the second subsection, we implement approximations to the RTRL gradient in order to overcome these undesirable features, leading to the RFLO learning rules.

In the third subsection of this appendix, we review the derivation of BPTT, the most widely used algorithm for training RNNs. Because it is the standard gradient-based learning rule for RNN training, BPTT is the learning rule against which we compare RFLO learning in the main text. Finally, in the final subsection of this appendix we illustrate the equivalence of RTRL and BPTT. Although this is not strictly necessary for any of the results given in the main text, we expect that readers with an interest in gradient-based learning rules for training RNNs will be interested in this correspondence, which to our knowledge has not been very clearly explicated in the literature.

## Real-time recurrent learning

In this section we review the derivation of the real-time recurrent learning (RTRL) algorithm (*Williams and Zipser, 1989*) for an RNN such as the one shown in *Figure 1*. This rule is obtained by taking a gradient of the mean-squared output error of the RNN with respect to the synaptic weights, and, as we will show later in this appendix, is equivalent (when implemented in batches rather than online) to the more widely used backpropagation through time (BPTT) algorithm.

The standard RTRL algorithm is obtained by calculating the gradient of the loss function *Equation (2)* with respect to the RNN weights, and then using gradient descent to find the weights that minimize the loss function (*Goodfellow et al., 2016*). Specifically, for each run of the network, one can calculate $\partial L/\partial W_{ab}$ and then update the weights by an amount proportional to this gradient: $\Delta W_{ab} = -\eta \partial L/\partial W_{ab}$, where $\eta$ determines the learning rate. This can be done similarly for the input and output weights, $W_{ab}^{\text{in}}$ and $W_{ab}^{\text{out}}$, respectively. This results in the following update equations:

$$\Delta W_{ab}^{\text{out}} = \frac{\eta_1}{T} \sum_{t=1}^{T} \varepsilon_a(t) h_b(t),$$

$$\Delta W_{ab} = \frac{\eta_2}{T} \sum_{t=1}^{T} \sum_{j=1}^{N} \left[ (\mathbf{W}^{\text{out}})^{\text{T}} \varepsilon(t) \right]_j \frac{\partial h_j(t)}{\partial W_{ab}},$$

$$\Delta W_{ab}^{\text{in}} = \frac{\eta_3}{T} \sum_{t=1}^{T} \sum_{j=1}^{N} \left[ (\mathbf{W}^{\text{out}})^{\text{T}} \varepsilon(t) \right]_j \frac{\partial h_j(t)}{\partial W_{ab}^{\text{in}}}. \tag{10}$$

In these equations, $(\cdot)^{\text{T}}$ denotes matrix transpose, and the gradients of the hidden layer activities with respect to the recurrent and input weights are given by

$$P_{ab}^j(t) = \left(1 - \frac{1}{\tau}\right)P_{ab}^j(t-1) + \frac{1}{\tau}\sum_k \phi'\left(u_j(t)\right)W_{jk}P_{ab}^k(t-1)$$

$$+ \frac{1}{\tau}\delta_{ja}\phi'(u_a(t))h_b(t-1),$$

$$Q_{ab}^j(t) = \left(1 - \frac{1}{\tau}\right)Q_{ab}^j(t-1) + \frac{1}{\tau}\sum_k \phi'\left(u_j(t)\right)W_{jk}Q_{ab}^k(t-1)$$

$$+ \frac{1}{\tau}\delta_{ja}\phi'(u_a(t))x_b(t-1), \tag{11}$$

where we have defined

$$P_{ab}^j(t) \equiv \frac{\partial h_j(t)}{\partial W_{ab}},$$

$$Q_{ab}^j(t) \equiv \frac{\partial h_j(t)}{\partial W_{ab}^{\text{in}}}, \tag{12}$$

and $\mathbf{u}(t)$ is the total input to each recurrent unit at time $t$:

$$u_i(t) = \sum_{j=1}^N W_{ij}h_j(t-1) + \sum_{j=1}^{N_x} W_{ij}^{\text{in}}x_j(t). \tag{13}$$

The recursions in *Equation (11)* terminate with

$$\frac{\partial h_j(0)}{\partial W_{ab}} = 0,$$

$$\frac{\partial h_j(0)}{\partial W_{ab}^{\text{in}}} = 0. \tag{14}$$

As many others have recognized previously, the synaptic weight updates given in the second and third lines of *Equation (10)* are not biologically realistic for a number of reasons. First, the error is projected back into the network with the particular weight matrix $(W^{\text{out}})^{\text{T}}$, so that the feedback and readout weights must be related to one another in a highly specific way. Second, the terms involving $\mathbf{W}$ in *Equation (11)* mean that information about the entire network is required to update any given synaptic weight, making the rules nonlocal. In contrast, a biologically plausible learning rule for updating a weight $W_{ab}$ or $W_{ab}^{\text{in}}$ ought to depend only on the activity levels of the pre- and post-synaptic units $a$ and $b$, in addition to the error signal that is fed back into the network. Both of these shortcomings will be addressed in the following subsection.

## Random feedback local online learning

In order to obtain a biologically plausible learning rule, we can attempt to relax some of the requirements in the RTRL learning rule and see whether the RNN is still able to learn effectively. Inspired by a recently used approach in feedforward networks (*Lillicrap et al., 2016*), we do this by replacing the $(W^{\text{out}})^{\text{T}}$ appearing in the second and third lines of *Equation (10)* with a fixed random matrix $\mathbf{B}$, so that the feedback projection of the output error no longer needs to be tuned to match the other weights in the network in a precise way. Second, we simply drop the terms involving $\mathbf{W}$ in *Equation (11)*, so that nonlocal information about all recurrent weights in the network is no longer required to update a particular synaptic weight. In this case we can rewrite the approximate weight-update equations as

$$\Delta W_{ab}^{\text{out}} = \frac{1}{T}\sum_{t=1}^{T} \delta W_{ab}^{\text{out}}(t),$$

$$\Delta W_{ab} = \frac{1}{T}\sum_{t=1}^{T} \delta W_{ab}(t),$$

$$\Delta W_{ab}^{\text{in}} = \frac{1}{T}\sum_{t=1}^{T} \delta W_{ab}^{\text{in}}(t), \tag{15}$$

where

$$\delta W_{ab}^{\text{out}}(t) = \eta_1 \varepsilon_a(t) h_b(t),$$

$$\delta W_{ab}(t) = \eta_2 [\mathbf{B}\varepsilon(t)]_a p_{ab}(t),$$

$$\delta W_{ab}^{\text{in}}(t) = \eta_3 [\mathbf{B}\varepsilon(t)]_a q_{ab}(t). \tag{16}$$

Here we have defined rank-2 versions of the eligibility trace tensors from (12):

$$p_{ab}(t) = \frac{1}{\tau}\phi'(u_a(t-1))h_b(t-1) + \left(1 - \frac{1}{\tau}\right)p_{ab}(t-1),$$

$$q_{ab}(t) = \frac{1}{\tau}\phi'(u_a(t-1))x_b(t-1) + \left(1 - \frac{1}{\tau}\right)q_{ab}(t-1). \tag{17}$$

As desired, the *Equation (15)* are local, depending only on the pre- and post-synaptic activity, together with a random feedback projection of the error signal. In addition, because all of the quantities appearing in *Equation (15)* are computed in real time as the RNN is run, the weight updates can be performed *online*, in contrast to BPTT, for which the dynamics over all timesteps must be run first forward and then backward before making any weight updates. Hence, we refer to the learning rule given by (15 - 12) as random feedback local online (RFLO) learning.

## Backpropagation through time

Because it is the standard algorithm used for training RNNs, in this section we review the derivation of the learning rules for backpropagation through time (BPTT) (*Rumelhart et al., 1985*) in order to compare it with the learning rules presented above. The derivation here follows *Lecun (1988)*.

Consider the following Lagrangian function:

$$\mathcal{L}\big[\mathbf{h},\mathbf{z},\mathbf{W},\mathbf{W}^{\text{in}},\mathbf{W}^{\text{out}},t\big] = \sum_i z_i(t)\left\{ h_i(t) - h_i(t-1) + \frac{1}{\tau}\Big[h_i(t-1) - \phi\Big(\big[\mathbf{W}\mathbf{h}(t-1) + \mathbf{W}^{\text{in}}\mathbf{x}(t)\big]_i\Big)\Big] \right\}$$

$$+ \frac{1}{2}\sum_i \Big(y_i^*(t) - \big[\mathbf{W}^{\text{out}}\mathbf{h}(t)\big]_i\Big)^2. \tag{18}$$

The second line is the cost function that is to be minimized, while the first line uses the Lagrange multiplier $\mathbf{z}(t)$ to enforce the constraint that the dynamics of the RNN should follow *Equation (1)*. From *Equation (18)* we can also define the following action:

$$S\big[\mathbf{h},\mathbf{z},\mathbf{W},\mathbf{W}^{\text{in}},\mathbf{W}^{\text{out}}\big] = \frac{1}{T}\sum_{t=1}^{T} \mathcal{L}\big[\mathbf{h},\mathbf{z},\mathbf{W},\mathbf{W}^{\text{in}},\mathbf{W}^{\text{out}},t\big]. \tag{19}$$

We now proceed by minimizing *Equation (19)* with respect to each of its arguments. First, taking $\partial S/\partial z_i(t)$ just gives the dynamical *Equation (1)*. Next, we set $\partial S/\partial h_i(t) = 0$, which yields

$$z_i(t) = \left(1 - \frac{1}{\tau}\right)z_i(t+1) + \frac{1}{\tau}\sum_j z_j(t+1)\phi'\Big(\big[\mathbf{W}\mathbf{h}(t) + \mathbf{W}^{\text{in}}\mathbf{x}(t+1)\big]_j\Big)W_{ji} + \Big[(W^{\text{out}})^{\text{T}}\varepsilon(t)\Big]_i, \tag{20}$$

which applies at timesteps $t = 1,\ldots,T-1$. To obtain the value at the final timestep, we take $\partial S/\partial h_i(T)$, which leads to

$$z_i(T) = \left[ (W^{\text{out}})^T \varepsilon(T) \right]_i.\tag{21}$$

Finally, taking the derivative with respect to the weights leads to the following:

$$\frac{\partial S}{\partial W_{ij}} = -\frac{1}{T\tau} \sum_{t=1}^{T} z_i(t) \phi' \left( \left[ \mathbf{W}\mathbf{h}(t-1) + \mathbf{W}^{\text{in}}\mathbf{x}(t) \right]_i \right) h_j(t-1)$$

$$\frac{\partial S}{\partial W_{ij}^{\text{in}}} = -\frac{1}{T\tau} \sum_{t=1}^{T} z_i(t) \phi' \left( \left[ \mathbf{W}\mathbf{h}(t-1) + \mathbf{W}^{\text{in}}\mathbf{x}(t) \right]_i \right) x_j(t)$$

$$\frac{\partial S}{\partial W_{ij}^{\text{out}}} = -\frac{1}{T} \sum_{t=1}^{T} \varepsilon_i(t) h_j(t).\tag{22}$$

Rather than setting these derivatives equal to zero, which may lead to an undesired solution that corresponds to a maximum or saddle point of the action and would in any case be intractable, we use the gradients in *Equation (22)* to perform gradient descent, reducing the error in an iterative fashion:

$$\Delta W_{ij} = \frac{\eta_2}{T\tau} \sum_{t=1}^{T} z_i(t) \phi' \left( \left[ \mathbf{W}\mathbf{h}(t-1) + \mathbf{W}^{\text{in}}\mathbf{x}(t) \right]_i \right) h_j(t-1)$$

$$\Delta W_{ij}^{\text{in}} = \frac{\eta_3}{T\tau} \sum_{t=1}^{T} z_i(t) \phi' \left( \left[ \mathbf{W}\mathbf{h}(t-1) + \mathbf{W}^{\text{in}}\mathbf{x}(t) \right]_i \right) x_j(t)$$

$$\Delta W_{ij}^{\text{out}} = \frac{\eta_1}{T} \sum_{t=1}^{T} \varepsilon_i(t) h_j(t),\tag{23}$$

where $\eta_i$ are learning rates.

The BPTT algorithm then proceeds in three steps. First, the dynamical *Equation (1)* for $\mathbf{h}(t)$ are integrated forward in time, beginning with the initial condition $\mathbf{h}(0)$. Second, the auxiliary variable $\mathbf{z}(t)$ is integrated *backwards* in time using *Equation (20)*, using with the $\mathbf{h}(t)$ saved from the forward pass and the boundary condition $\mathbf{z}(T)$ from *Equation (21)*. Third, the weights are updated according to *Equation (23)*, using $\mathbf{h}(t)$ and $\mathbf{z}(t)$ saved from the preceding two steps.

Note that no approximations have been made in computing the gradients using either the RTRL or BPTT procedures. In fact, as we will show in the following section, the two algorithms are completely equivalent, at least in the case where RFLO weight updates are performed only at the end of each trial rather than at every timestep.

## A unified view of gradient-based learning in recurrent networks

As pointed out previously (*Beaufays and Wan, 1994*; *Srinivasan et al., 1994*), the difference between RTRL and BPTT can ultimately be traced to distinct methods of bookkeeping in applying the chain rule to the gradient of the loss function. (Thanks to A. Litwin-Kumar for discussion about this correspondence). In order to make this explicit, we begin by noting that, when taking implicit dependences into account, the loss function defined in *Equation (2)* has the form

$$L = L\left( \mathbf{h}^0, \ldots, \mathbf{h}^t \left( \mathbf{W}, \mathbf{h}^{t-1} \left( \mathbf{W}, \mathbf{h}^{t-2}(\ldots) \right) \right), \ldots \right).\tag{24}$$

In this section, we write $\mathbf{h}^t \equiv \mathbf{h}(t)$ for notational convenience, and consider only updates to the recurrent weights $\mathbf{W}$, ignoring the input $\mathbf{x}(t)$ to the RNN. In any gradient-based learning scheme, the weight update $\Delta W_{ab}$ should be proportional to the gradient of the loss function, which has the form

$$\frac{\partial L}{\partial W_{ab}} = \sum_t \frac{\partial L}{\partial \mathbf{h}^t} \cdot \frac{\partial \mathbf{h}^t}{\partial W^{ab}}. \tag{25}$$

The difference between RTRL and BPTT arises from the two possible ways of keeping track of the implicit dependencies from *Equation (24)*, which give rise to the following equivalent formulations of *Equation (25)*:

$$\frac{\partial L}{\partial W_{ab}} = \begin{cases} \sum_t \frac{\partial L(\dots,\mathbf{h}^t,\dots)}{\partial \mathbf{h}^t} \cdot \frac{\partial \mathbf{h}^t(\mathbf{W},\mathbf{h}^{t-1}(\mathbf{W},\mathbf{h}^{t-2}(\dots)))}{\partial W_{ab}}, & \text{RTRL} \\ \sum_t \frac{\partial L\left(\dots,\mathbf{h}^t\left(\mathbf{W},\mathbf{h}^{t-1}(\dots)\right),\mathbf{h}^{t+1}(\mathbf{W},\mathbf{h}^t(\dots)),\dots\right)}{\partial \mathbf{h}^t} \cdot \frac{\partial \mathbf{h}^t(\mathbf{W},\mathbf{h}^{t-1})}{\partial W_{ab}}. & \text{BPTT} \end{cases} \tag{26}$$

In RTRL, the first derivative is simple to compute because loss function is treated as an explicit function of the variables $\mathbf{h}^t$. The dependence of $\mathbf{h}^t$ on $\mathbf{W}$ and $\mathbf{h}^{t'}$ (where $t' < t$) is then taken into account in the second derivative, which must be computed recursively due to the nested dependence on $\mathbf{W}$. In BPTT, on the other hand, the implicit dependencies are dealt with in the first derivative, which in this case must be computed recursively because all terms at times $t' > t$ depend implicitly on $\mathbf{h}^t$. The second derivative then becomes simple since these dependencies are no longer present.

Let us define the following:

$$P^i_{ab}(t) \equiv \frac{\partial h^t_i(\mathbf{W},\mathbf{h}^{t-1}(\mathbf{W},\mathbf{h}^{t-2}(\dots)))}{\partial W_{ab}},$$

$$z_i(t) \equiv -\frac{\partial L\left(\dots,\mathbf{h}^t\left(\mathbf{W},\mathbf{h}^{t-1}\left(\mathbf{W},\mathbf{h}^{t-2}(\dots)\right)\right),\dots\right)}{\partial h^t_i}. \tag{27}$$

Then, using the definition of $L$ from *Equation (2)* and the dynamical *Equation (1)* for $\mathbf{h}^t$ to take the other derivatives appearing in *Equation (26)*, we have

$$\frac{\partial L}{\partial W_{ab}} = \begin{cases} -1T \sum_t \sum_i \left[(W^{\text{out}})^T \varepsilon(t)\right]_i P^i_{ab}(t), & \text{RTRL} \\ -1\tau T \sum_t z_a(t) \phi'(u_a(t)) h^{t-1}_b. & \text{BPTT} \end{cases} \tag{28}$$

The recursion relations follow from application of the chain rule in the definitions from *Equation (27)*:

$$P^i_{ab}(t) = \left(1 - \frac{1}{\tau}\right) P^i_{ab}(t-1) + \frac{1}{\tau} \sum_j \phi'(u_i(t-1)) W_{ij} P^j_{ab}(t-1) + \frac{1}{\tau} \delta_{ia} \phi'(u_a(t)) h_b(t-1),$$

$$z_i(t) = \left(1 - \frac{1}{\tau}\right) z_i(t+1) + \frac{1}{\tau} \sum_j \phi'(u_j(t+1)) W_{ji} z_j(t+1) + \sum_j W^{\text{out}}_{ji} \varepsilon_j(t). \tag{29}$$

These recursion relations are identical to those appearing in *Equation (11)* and *Equation (20)*. Notably, the first is computed forward in time, while the second is computed backward in time. Because no approximations have been made in computing the gradient in either case for *Equation (28)*, the two methods are equivalent, at least if RTRL weight updates are made only at the end of each trial, rather than online. For this reason, only one of the algorithms (BPTT) was compared against RFLO learning in the main text.

As discussed in previous sections, RTRL has the advantages of obeying causality and of allowing for weights to be continuously updated. But, as discussed above, RTRL has the disadvantage of being nonlocal, and also features a greater computational cost due to the necessity of updating a rank-3 tensor $P^i_{ab}(t)$ rather than a vector $z_i(t)$ at each timestep. By dropping the second term in the first line of *Equation (29)*, RFLO learning eliminates both of these undesirable features, so that the resulting algorithm is causal, online, local, and has a computational complexity ($\sim N^2$ per timestep, vs. $\sim N^4$ for RTRL) on par with BPTT.

## Appendix 2

DOI: https://doi.org/10.7554/eLife.43299.011

### Analysis of the RFLO learning rule

Given that the learning rules in *Equation (7)* do not move the weights directly along the steepest path that would minimize the loss function (as would the learning rules in *Equation (10)*), it is worthwhile to ask whether it can be shown that these learning rules in general decrease the loss function at all. To answer this question, we consider the change in weights after one trial lasting $T$ timesteps, working in the continuous-time limit for convenience, and performing weight updates only at the end of the trial:

$$\Delta\mathbf{W} = \frac{1}{T}\int_0^T dt\,\delta\mathbf{W}(t),$$

$$\Delta\mathbf{W}^{\text{out}} = \frac{1}{T}\int_0^T dt\,\delta\mathbf{W}^{\text{out}}(t) \tag{30}$$

where $\delta\mathbf{W}$ and $\delta\mathbf{W}^{\text{out}}$ are given by *Equation (7)*. For simplicity in this section we ignore the updates to the input weights, since the results in this case are very similar to those for recurrent weight updates.

In the first subsection of this appendix, we show that, under some approximations, the loss function tends to decrease on average under RFLO learning if there is positive alignment between the readout weights $\mathbf{W}^{\text{out}}$ and the feedback weights $\mathbf{B}$. In the second subsection, we show that this alignment tends to increase during RFLO learning.

### Decrease of the loss function

We first consider the change in the loss function defined in *Equation (2)* after updating the weights:

$$\Delta L = \frac{1}{2T}\int_0^T dt\left[\varepsilon^2(t,\mathbf{W}+\Delta\mathbf{W},\mathbf{W}^{\text{out}}+\Delta\mathbf{W}^{\text{out}}) - \varepsilon^2(t,\mathbf{W},\mathbf{W}^{\text{out}})\right]. \tag{31}$$

Assuming the weight updates to be small, we ignore terms beyond leading order in $\Delta\mathbf{W}$ and $\Delta\mathbf{W}^{\text{out}}$. Then, using the update rules in *Equation (30)* and performing some algebra, *Equation (31)* becomes

$$\Delta L = -\frac{\eta_1}{T}\sum_{ab}\left[\int_0^T \frac{dt}{T}\varepsilon_a(t)h_b(t)\right]^2$$

$$- \frac{\eta_2}{T}\int_0^T \frac{dt}{T}\int_0^T \frac{dt'}{T}\sum_{ab}\sum_{ijk}W_{ij}^{\text{out}}B_{ak}\varepsilon_i(t)P_{ab}^j(t)\varepsilon_k(t')p_{ab}(t')$$

$$\equiv \Delta L^{(1)} + \Delta L^{(2)}. \tag{32}$$

Clearly the first term in *Equation (32)* always tends to decrease the loss function, as we would expect given that the precise gradient of $L$ with respect to $\mathbf{W}^{\text{out}}$ was used to determine this part of the learning rule. We now wish to show that, at least on average and with some simplifying assumptions, the second term in *Equation (32)* tends to be negative as well. Before beginning, we note in passing that this term is manifestly nonpositive like the first term if we perform RTRL, in which case $\sum_k B_{ak}\varepsilon_k(t')p_{ab}(t') \rightarrow \sum_{kl} W_{kl}^{out}\varepsilon_k(t')P_{ab}^l(t')$ in *Equation (32)*, making the gradient exact.

In order to analyze $\Delta L^{(2)}$, we will assume that the RNN is linear, with $\phi(x) = x$. Further, we will average over the RNN activity $\mathbf{h}(t)$, assuming that the activities are correlated from one timestep to the next, but not from one unit to the next:

$$\langle h_i(t) h_j(t') \rangle_{\mathrm{h}} = \delta_{ij} C(t - t'). \tag{33}$$

The correlation function should be peaked at a positive value at $t - t' = 0$ and decay to 0 at much earlier and later times. Finally, because of the antisymmetry under $x \to -x$, odd powers of $\mathbf{h}$ will average to zero: $\langle h_i \rangle_{\mathrm{h}} = \langle h_i h_j h_k \rangle_{\mathrm{h}} = 0$.

With these assumptions, we can express the activity-averaged second line of *Equation (32)* as $\langle \Delta L^{(2)} \rangle_{\mathrm{h}} = F_1 + F_2$, with

$$F_1 = -\frac{\eta_2}{T} N \sum_{ajkl} W_{kj}^{\mathrm{out}} B_{al} \int_0^T \frac{dt}{T} \int_0^T \frac{dt'}{T} y_k^*(t) y_l^*(t')$$
$$\times \int_0^t \frac{du}{\tau} \left[ e^{(1-\mathbf{W})(u-t)/\tau} \right]_{ja} \int_0^{t'} \frac{du'}{\tau} e^{-u'/\tau} C(t' - u' - u), \tag{34}$$

and

$$F_2 = -\frac{\eta_2}{T} N \sum_{ajklm} W_{kj}^{\mathrm{out}} W_{kl}^{\mathrm{out}} B_{am} W_{ml}^{\mathrm{out}} \int_0^T \frac{dt}{T} \int_0^T \frac{dt'}{T} \int_0^t \frac{du}{\tau} \int_0^{t'} \frac{du'}{\tau}$$
$$\times e^{-u'/\tau} \left[ e^{(1-\mathbf{W})(u-t)/\tau} \right]_{ja} C(t - t') C(t' - u' - u) + O(N^0). \tag{35}$$

In order to make further progress, we can perform an ensemble average over $\mathbf{W}$, assuming that $W_{ij} \sim \mathcal{N}(0, g^2/N)$ is a random variable, which leads to

$$\left\langle \left[ e^{(1-\mathbf{W})(u-t)/\tau} \right]_{ja} \right\rangle_{\mathbf{W}} = \delta_{ja} e^{(u-t)/\tau} + O(g^2/N). \tag{36}$$

This leads to

$$F_1 = -\frac{\eta_2}{T} N \int_0^T \frac{dt}{T} \int_0^T \frac{dt'}{T} [\mathbf{y}^*(t)]^{\mathrm{T}} \mathbf{W}^{\mathrm{out}} \mathbf{B} \mathbf{y}^*(t')$$
$$\times \int_0^t \frac{du}{\tau} \int_0^{t'} \frac{du'}{\tau} e^{(-t+u-u')/\tau} C(t' - u' - u) + O(N^0), \tag{37}$$

and

$$F_2 = -\frac{\eta_2}{T} N \mathrm{Tr} \left[ (\mathbf{W}^{\mathrm{out}})^{\mathrm{T}} \mathbf{W}^{\mathrm{out}} \mathbf{B} \mathbf{W}^{\mathrm{out}} \right] \int_0^T \frac{dt}{T} \int_0^T \frac{dt'}{T} \int_0^t \frac{du}{\tau} \int_0^{t'} \frac{du'}{\tau}$$
$$\times e^{(-t+u-u')/\tau} C(t - t') C(t' - u' - u) + O(N^0). \tag{38}$$

Putting *Equation (37)* and *Equation (38)* together, changing one integration variable, and dropping the terms smaller than $O(N)$ then gives

$$\langle \Delta L^{(2)} \rangle_{\mathrm{h}, \mathbf{W}} = -\frac{\eta_2}{T} N \int_0^T \frac{dt}{T} \int_0^T \frac{dt'}{T} \int_0^t \frac{du}{\tau} \int_0^{t'} \frac{dv}{\tau} e^{(-t-t'+u+v)/\tau} C(u-v)$$
$$\times \left\{ [\mathbf{y}^*(t)]^{\mathrm{T}} \mathbf{W}^{\mathrm{out}} \mathbf{B} \mathbf{y}^*(t') + C(t - t') \mathrm{Tr} \left[ (\mathbf{W}^{\mathrm{out}})^{\mathrm{T}} \mathbf{W}^{\mathrm{out}} \mathbf{B} \mathbf{W}^{\mathrm{out}} \right] \right\}. \tag{39}$$

Because we have assumed that $C(t) \geq 0$, the sign of this quantity depends only on the sign of the two terms in the second line of *Equation (39)*.

Already we can see that *Equation (39)* will tend to be negative when $\mathbf{W}^{\mathrm{out}}$ is aligned with $B$. To see this, suppose that $\mathbf{B} = \alpha \mathbf{W}^{\mathrm{out}}$, with $\alpha > 0$. Due to the exponential factor, the integrand will be vanishingly small except when $t \approx t'$, so that the first term in the second line in this case can be written as $\approx \alpha |(\mathbf{W}^{\mathrm{out}})^{\mathrm{T}} \mathbf{y}^*(t)|^2 \geq 0$. The second term, meanwhile, becomes $\alpha C(t - t') \mathrm{Tr} \left[ ((\mathbf{W}^{\mathrm{out}})^{\mathrm{T}} \mathbf{W}^{\mathrm{out}})^2 \right] \geq 0$.

The situation is most transparent if we assume that the RNN readout is one-dimensional, in which case the readout and feedback weights become vectors $\mathbf{w}^{\mathrm{out}}$ and $\mathbf{b}$, respectively, and *Equation (39)* becomes

$$\langle \Delta L^{(2)}\rangle_{\mathbf{h},\mathbf{W}} = -\frac{\eta_2}{T}N\int_0^T\frac{dt}{T}\int_0^T\frac{dt'}{T}\int_0^t\frac{du}{\tau}\int_0^{t'}\frac{dv}{\tau}e^{(-t-t'+u+v)/\tau}C(u-v)$$
$$\times\left\{y^*(t)y^*(t')\mathbf{w}^{\text{out}}\cdot\mathbf{b} + C(t-t')|\mathbf{w}^{\text{out}}|^2\mathbf{w}^{\text{out}}\cdot\mathbf{b}\right\}. \tag{40}$$

In this case it is clear that, as in the case of feedforward networks (*Lillicrap et al., 2016*), the loss function tends to decrease when the readout weights become aligned with the feedback weights. In the following subsection we will show that, at least under similar approximations to the ones made here, such alignment does in fact occur.

## Alignment of readout weights with feedback weights

In the preceding subsection it was shown that, assuming a linear RNN and averaging over activities and recurrent weights, the loss function tends to decrease when the alignment between the readout weights $\mathbf{W}^{\text{out}}$ and the feedback weights $\mathbf{B}$ becomes positive. In this subsection we ask whether such alignment does indeed occur.

In order to address this question, we consider the quantity $\text{Tr}(\mathbf{W}^{\text{out}}\mathbf{B})$ and ask how it changes following one cycle of training, with combined weight updates on $\mathbf{W}$ and $\mathbf{W}^{\text{out}}$. (As in the preceding subsection, external input to the RNN is ignored here for simplicity.) The effect of modifying the readout weights is obvious from *Equation (15)*:

$$\Delta\text{Tr}(\mathbf{W}^{\text{out}}\mathbf{B}) = \text{Tr}((\Delta\mathbf{W}^{\text{out}})\mathbf{B})$$
$$= \eta_1\sum_{ab}B_{ba}\int_0^T\frac{dt}{T}\varepsilon_a(t)h_b(t). \tag{41}$$

The update to the recurrent weights, on the other hand, modifies $\mathbf{h}(t)$ in the above equation. Because we are interested in the combined effect of the two weight updates and are free to make the learning rates arbitrarily small, we focus on the following quantity:

$$G \equiv \frac{\partial^2}{\partial\eta_1\partial\eta_2}\Big|_{\eta_1,\eta_2=0}\Delta\text{Tr}(\mathbf{W}^{\text{out}}\mathbf{B})$$
$$= \frac{\partial}{\partial\eta_2}\Big|_{\eta_2=0}\sum_{ab}B_{ba}\int_0^T\frac{dt}{T}\varepsilon_a(t)h_b(t). \tag{42}$$

The goal of this subsection is thus to show that (at least on average) $G>0$.

In order to evaluate this quantity, we need to know how the RNN activity $\mathbf{h}(t)$ depends on the weight modification $\Delta\mathbf{W}$. As in the preceding subsection, we will assume a linear RNN and will work in the continuous-time limit ($\tau \gg 1$) for convenience. In this case, the dynamics are given by

$$\tau\frac{d}{dt}\mathbf{h}(t) = (\mathbf{W}+\Delta\mathbf{W})\mathbf{h}(t). \tag{43}$$

If we wish to integrate this equation to get $\mathbf{h}(t)$ and expand to leading order in $\Delta\mathbf{W}$, care must be taken due to the fact that $\mathbf{W}$ and $\Delta\mathbf{W}$ are non-commuting matrices. Taking a cue from perturbation theory in quantum mechanics (*Sakurai, 1994*), we can work in the 'interaction picture' and obtain

$$\mathbf{h}(t) = e^{\mathbf{W}t/\tau}e^{\Delta\hat{\mathbf{W}}t/\tau}\mathbf{h}(0), \tag{44}$$

where

$$\Delta\hat{\mathbf{W}} \equiv e^{-\mathbf{W}t/\tau}\Delta\mathbf{W}e^{\mathbf{W}t/\tau}. \tag{45}$$

We can now expand *Equation (44)* to obtain

$$\mathbf{h}(t) = \left[ e^{\mathbf{W}t/\tau} + \frac{t}{\tau}\Delta\mathbf{W}e^{\mathbf{W}t/\tau} + O(\eta_2^2) \right]\mathbf{h}(0). \tag{46}$$

For a linear network, the update rule for $\mathbf{W}$ from **Equation (15)** is then simply

$$\Delta W_{ab} = \eta_2 \int_0^T \frac{dt}{T}\sum_c B_{ac}\varepsilon_c(t)\bar{h}_b(t), \tag{47}$$

where the bar denotes low-pass filtering:

$$\bar{\mathbf{h}}(t) = \int_0^t \frac{dt'}{\tau}e^{-t'/\tau}\mathbf{h}(t-t'). \tag{48}$$

Combining (**Equations (46–48)**), the time-dependent activity vector to leading order in $\eta_2$ is

$$h_i(t) = \hat{h}_i(t) + \eta_2\frac{t}{\tau}\sum_{jk}B_{ik}\int_0^T \frac{dt'}{T}\int_0^{t'}\frac{dt''}{\tau}e^{-t''/\tau}\left[y_k^*(t') - \sum_l W_{kl}^{\text{out}}\hat{h}_l(t')\right]\hat{h}_j(t'-t'')\hat{h}_j(t), \tag{49}$$

where $\hat{\mathbf{h}}(t)$ is the unperturbed RNN activity vector (i.e. without the weight update $\Delta\mathbf{W}$). With this result, we can express **Equation (42)** as $G = G_1 + G_2$, where

$$G_1 = \sum_{ab}\sum_{ij}B_{ba}B_{bj}\int_0^T \frac{dt}{T}\int_0^T \frac{dt'}{T}\frac{t}{\tau}\int_0^{t'}\frac{dt''}{\tau}e^{-t''/\tau}\hat{\varepsilon}_a(t)\hat{\varepsilon}_j(t')\hat{h}_i(t)\hat{h}_i(t'-t'') \tag{50}$$

and

$$G_2 = -\sum_{ab}\sum_{ijk}B_{ba}B_{ik}W_{ai}^{\text{out}}\int_0^T \frac{dt}{T}\int_0^T \frac{dt'}{T}\frac{t}{\tau}\int_0^{t'}\frac{dt''}{\tau}e^{-t''/\tau}\hat{h}_b(t)\hat{\varepsilon}_k(t')\hat{h}_j(t)\hat{h}_j(t'-t''). \tag{51}$$

Here we have defined $\hat{\varepsilon}(t) \equiv \mathbf{y}^*(t) - \mathbf{W}^{\text{out}}\hat{\mathbf{h}}(t)$.

In order to make further progress, we follow the approach of the previous subsection and perform an average over RNN activity vectors, which yields

$$\langle G_1\rangle_{\hat{\mathbf{h}}} = \frac{N}{\tau}\int_0^T \frac{dt}{T}t\int_0^T \frac{dt'}{T}\left\{\mathbf{y}^*(t)\cdot\mathbf{B}^{\mathrm{T}}\mathbf{B}\cdot\mathbf{y}^*(t')\int_0^{t'}\frac{dt''}{\tau}e^{-t''/\tau}C(t-t'+t'')\right.$$
$$\left. + \mathrm{Tr}\left[\mathbf{B}^{\mathrm{T}}\mathbf{B}\mathbf{W}^{\text{out}}(\mathbf{W}^{\text{out}})^{\mathrm{T}}\right]\int_0^{t'}\frac{dt''}{\tau}e^{-t''/\tau}C(t-t')C(t-t'+t'') + O(1/N)\right\} \tag{52}$$

and

$$\langle G_2\rangle_{\hat{\mathbf{h}}} = \frac{N}{\tau}\mathrm{Tr}(\mathbf{B}\mathbf{W}^{\text{out}}\mathbf{B}\mathbf{W}^{\text{out}})\int_0^T \frac{dt}{T}t\int_0^T \frac{dt'}{T}\int_0^{t'}\frac{dt''}{\tau}e^{-t''/\tau}[C(t-t')C(t-t'+t'') + O(1/N)]. \tag{53}$$

Similar to the integral in **Equation (39)**, both of these quantities will tend to be positive if we assume that $C(t) \geq 0$ with a peak at $t = 0$, and note that the integrand is large only when $t \approx t'$.

In order to make the result even more transparent, we can again consider the case of a one-dimensional readout, in which case **Equation (52)** becomes

$$\langle G_1\rangle_{\hat{\mathbf{h}}} = \frac{N|\mathbf{b}|^2}{\tau}\int_0^T \frac{dt}{T}t\int_0^T \frac{dt'}{T}\int_0^{t'}\frac{dt''}{\tau}e^{-t''/\tau}[y^*(t)y^*(t')C(t-t'+t'')$$
$$+ |\mathbf{w}^{\text{out}}|^2C(t-t')C(t-t'+t'') + O(1/N)] \tag{54}$$

and

$$\langle G_2 \rangle_{\hat{\mathbf{h}}} = \frac{N}{\tau}(\mathbf{w}^{\text{out}} \cdot \mathbf{b})^2 \int_0^T \frac{dt}{T} t \int_0^T \frac{dt'}{T} \int_0^{t'} \frac{dt''}{\tau} e^{-t''/\tau}[C(t-t')C(t-t'+t'') + O(1/N)] \qquad (55)$$

This version illustrates even more clearly that the right hand sides of these equations tend to be positive.

*Equation (52)* (or, in the case of one-dimensional readout, *Equation (54)*) shows that the overlap between the readout weights and feedback weights tends to increase with training. *Equation (39)* (or *Equation (40)*) then shows that the readout error will tend to decrease during training given that this overlap is positive. While these mathematical results provide a compelling plausibility argument for the efficacy of RFLO learning, it is important to recall that some limiting assumptions were required in order to obtain them. Specifically, we assumed linearity of the RNN and vanishing of the cross-correlations in the RNN activity, neither of which is strictly true in a trained nonlinear network. In order to show that RFLO learning remains effective even without these limitations, we must turn to numerical simulations such as those performed in the main text.

