## [Decision Letter]

Thank you for submitting your article "Local online learning in recurrent networks with random feedback" for consideration by *eLife*. Your article has been reviewed by three peer reviewers, one of whom is a member of our Board of Reviewing Editors, and the evaluation has been overseen by Michael Frank as the Senior Editor. The following individual involved in review of your submission has agreed to reveal their identity: Brian DePasquale (Reviewer #2).

The reviewers have discussed the reviews with one another and the Reviewing Editor has drafted this decision to help you prepare a revised submission.

The manuscript develops a new algorithm for training recurrent neural networks (RFLO). The algorithm is intended to be local and online, and uses random feedback connections to send error into the network. The algorithm is presented as a contribution to a growing set of work on how learning in neural networks might take place in the brain. The results demonstrate on a series of simple tasks that RFLO works about as well as methods that use full gradient information (e.g. BPTT and RTRL) in cases where the timescale of the task is relatively short. In cases where the timescales are longer, RFLO struggles versus BPTT. The manuscript makes some effort to explore what component of RFLO damages performance and suggests that the difficulties with long timescales may be dealt with via different mechanisms all together (e.g. by stitching together multiple shorter behaviours).

We found the paper well written and have a few essential revisions.

Some of our points are really suggestions that we hope will improve the paper. However, that's always a matter of opinion, so feel free to ignore us on those. We'll be specific by putting (suggestion) or (strong suggestion), the latter for the ones we think are important, before our comments.

Essential revisions:

1) (strong suggestion) The distinct modifications to BPTT/RTRL and the consequences of these modifications should be amplified in the existing text, to ensure that readers are sure to appreciate the results. For readers that are not familiar with gradient-based learning in RNNs and the ideas of random feedback, I fear that the separate ideas of gradient approximation and of random feedback might be muddled. Stressing that these two ideas are only related insofar as they separately address distinct shortcomings of existing algorithms would stress that they were introduced by the author not because they share a special relationship, but only to achieve that goal. This came across somewhat in the text, but I believe it could be amplified. Additionally, stressing earlier in the text why exactly finely-tuned feedback is a problem would be helpful (this might not be obvious to everyone). It's touched on in Discussion but explaining the concern in the Introduction would help frame that result more clearly.

2) (strong suggestion) Much of the math is relegated to the appendix. This seems like a pity since 1) the relationship of RFLO to BPTT is not obvious from the main text (undoubtedly requiring the reader to visit the appendix) and 2) the connection between RTRL and BPTT is presented so clearly (and, as the author noted, this connection has not been presented clearly in the literature). At a minimum, I would recommend migrating a minimal amount of the appendix to the main text to illustrate how RFLO relates to BPTT and RTRL. (Maybe, for example, Equation 9 could be presented in the main text). The tradeoff between BPTT and RTRL (trading-off non-locality and causality) and how RFLO addresses these non-biological features can only really be understood within the appendix, and I fear that not all readers will invest in it.

3) Figure 2B seems to indicate that performance decreases consistently as the period grows longer and then perhaps asymptotes. It would be nice to know how performance for the RFLO changes as a function of the period, over a broader range of values (at least for this particular periodic task), to understand a reasonable timescale under which learning could occur. This is directly relevant to the author's proposal about how the brain might learn (i.e. through a sequence of actions) because it will dictate the duration of a "behavioral syllable" based on this learning rule.

Other points:

1) The theory showing that the learning rule decreases the error makes a number of assumptions: linear dynamics, uncorrelated neurons, and random weights. This seems very problematic: linear dynamics correlates the neurons, and the whole point of learning is to make the weights non-random (and presumably, correlated). Thus, it's not clear what the theory adds. At the very least, this should be pointed out in the main text. Even better (but not required) would be to try to verify, numerically, some of the expressions in the Appendix. That won't be particularly easy, but it seems possible. Alternatively, the change in error versus the alignment, and the alignment versus time, could be plotted during a simulation. This would go a long way toward supporting – or refuting – the theory.

2) We couldn't find what the initial weights were in the learning rules. It would be good to know if small initial weights were needed, or if the learning rule works when the initial weights are large enough that the network is in the chaotic regime. (suggestion) In particular, a plot of performance versus initial weights (presumably the variance) would be informative.

3) In our experience, for local learning rules the variance in the error is large. The variance should be reported – not just the mean. In addition, more than 5 networks should be used to compute the error.

4) The networks were small (30 in Figure 2; 100 in Figure 3; not sure in Figure 4). What happens when the size increases? We're hesitant to ask for more simulations. However, if simulations with larger networks are not done, you need to be upfront about the fact that this study may not scale well to large networks.

5) (Very important!) The Materials and methods section should contain all simulation details. As far as we can tell, some are missing: the initial values of the weights, the learning rates (after the grid search), explicit forms for the target functions in Figure 2, and the time step. And we may have missed other details; you should make sure that there's enough information that the simulations can be replicated. It's true that the code is supplied, but not all of us like to read code.

6) You should also discuss Hoerzer et al. (2012, Cerebral cortex, 24(3), 677-690). This is an example of node perturbation without noise; it's instead based on overall performance relative to a running average. In that paper they train only the output weights, not the recurrent weights as is done here. However, if the network can do better than a training paradigm involving recurrent weights, it's worth mentioning. (suggestion) We would even go so far as to suggest comparing performance when training only the output weight against performance when training the recurrent weights.

7) The shaded regions in Figure 2 are only explained in the caption for panel D. It would be helpful to explain them in panel A as well.

8) The symbol 'i' is used to index over neurons in equation 1 (top), over outputs in the lower equation of equation 1 (and thus over the outputs in equation 2) and used to refer to the learning rates in the first sentence after equation 3. Given how technical the indexing can become, we would strongly suggest reserving 'i' (as you have for 'j','a', and 'b') for neurons only.

9) The gray line in Figure 2B is lost in the text describing what each color represents. It should be enlarged and made more prominent.

10) It would be helpful to add a more intuitive plotting convention for Figure 2C (such as a color gradient as τ gets longer).

11) (suggestion) In general, presenting the RFLO+Dale results in Figure 2 can be distracting from the main point (and the author doesn't treat it extensively in the text). We would suggest moving the Dale's Law results to a supplement to Figure 2 (see eLife's treatment of figure supplements).

12) The last line of text before equation 5 seems to contain incorrect references to equations 14 and 15 of the appendix, which I believe are the same as equations 3 and 4 of the main text.

13) Figure 2D appears to have a color mismatch between the line and the explanatory text within the figure, for "Local only".

14) The last sentence of "Interval matching" states that you will return to a point in the following subsection, but it's not clear which subsection you are referring to, or if that thread is ever discussed again later.

15) The algorithm is only run on simple toy problems that are constructed for the manuscript. The experiments hint that RFLO struggles in the context of longer timescales, but the manuscript provides no grounding for how well the algorithm performs on richer data. It is important to see performance of the algorithm gauged on a commonly used problem from the machine learning literature. Various datasets could be used, but the language modeling task on the Penn TreeBank (PTB) is very well explored and serves as a kind of MNIST for sequence modeling. Here it is important to quantify success in a fashion that is congruent with current standards in ML.

To be clear, high-level performance on such a task does not seem crucial. The paper is aimed at biology, and there is no reason to pursue top results. But it is important to be able to situate the obtained performance, and offer a benchmark for subsequent work on biologically plausible algorithms that simultaneously aim to be practical/functional.

16) (weak suggestion, since this will be tough, and possibly beyond the scope of the work. but it would be nice if it could be done) Along these lines, and in the context of an externally defined problem, it would be ideal to see the performance of the algorithm explored using an LSTM architecture. Basic RNNs performance tends to be quite poor relative to LSTMs across many tasks. Is it easy to adapt the RFLO algorithm to more complex architectures, and does doing so deliver better performance on any task? The answer may simply be no; if so, that's OK.

17) (weak suggestion) On this question of architecture: how well does RFLO function in the case that there are multiple 'layers' in the RNN (e.g. as in Deep LSTMs where the connectivity matrix is not all-to-all). Does the algorithm still function about as well, or does increasing the depth of the network slow training?

18) More could be done to emphasize that the tasks solved go well beyond what is solvable using a feedforward network and feedback alignment. This is implicit in some of the tasks, but guiding the reader to see this clearly and conclusively would be ideal.

19) In the original manuscript describing feedback alignment, convergence of error is proved under some very restrictive conditions. Please describe briefly, in the main text, how the theoretical results developed here are related to the original FA results. e.g. To what extend can they be seen as a generalization of those results? What assumptions are made differently or in addition to the FA results?

20) The manuscript briefly makes connections to the FORCE training method, but my feeling is that this currently doesn't go far enough. A few more sentences that summon more of the details of the model/algorithm from Sussillo and delineate the connections to RFLO would be useful to the reader.

21) (suggestion) The section on 'Learning a sequence of actions' is interesting, but currently feels ad-hoc. The section almost feels more like a long discussion point than something that ought to sit in the results. The message is, I believe, that: RFLO and similar algorithms may suffer relative to ML approaches such as BPTT on longer time scales, but this is ok because there are ways to rescue performance for long sequences. In particular, winner-take-all and reward modulated Hebbian learning rules are introduced along with additional sets of neurons to rescue performance on a movement sequence task. There is a brief attempt to relate these to the literature on structures that connect to motor cortex, but this feels rushed. Ideally, the manuscript would develop this section further so that it can be appreciated both with respect to biology and ML. Additionally, on this note, it would be good to see the performance of BPTT on the full sequence problem without these ad-hoc approaches. There is certainly a limit to what BPTT can do: how much does is it struggle with this situation? Is the long sequence one the BPTT also struggles with? This would provide grounding for where RFLO and these additional ideas sit with respect to BPTT training in this more interesting case.

22) There are a couple of citations that might be useful to include. For example, a mention of spiking variants of feedback alignment (e.g. "Deep learning with dynamic spiking neurons and fixed feedback weights"), and several recent works on approximations of RTRL in the ML literature that seem worth mentioning (e.g. "Unbiased online recurrent optimization", and "Approximating Real-Time Recurrent Learning with Random Kronecker Factors").

[Editors' note: further revisions were requested prior to acceptance, as described below.]

Thank you for submitting your article "Local online learning in recurrent networks with random feedback" for consideration by *eLife*. Your article has been reviewed by three peer reviewers, one of whom is a member of our Board of Reviewing Editors, and the evaluation has been overseen by Michael Frank as the Senior Editor. The following individual involved in review of your submission has agreed to reveal their identity: Brian DePasquale (Reviewer #2).

The reviewers have discussed the reviews with one another and the Reviewing Editor has drafted this decision to help you prepare a revised submission.

Summary:

This paper is essentially in, but there are two major (but not too hard) things left to do.

Essential revisions:

1) The treatment of Hoerzer et al. (a paper where you trained only the output weights) needs to be expanded. Our question last time was whether performance when only the output weights are trained could match performance when recurrent weights are trained. To address this, you trained only the output weights. However, as far as we could tell, there was no feedback. For a fair comparison, feedback weights are critical. Or at the very least, you should make sure you have a good weight initialization. (It is well known in the echo-state literature that when training only the output weights, the initialization of the other weight matrices is very important. The literature on echo-state networks contains lots of advice on how to ensure that these are set appropriately.) It would be nice if you included feedback weights and re-ran the simulations. That's not absolutely necessary, but if it's not done, then you will have to point out that you can't rule out the possibility that training the output weights actually works better than your approach of training the recurrent weights.

2) It would be good to include your response to concern 15 (attempting RFLO learning on more complicated problems) in the Discussion of the submitted manuscript. Your response at present is satisfactory, but the insight you share into why extending RFLO learning to more complex problems is itself interesting, and will likely be interesting to readers of the paper.

---

## [Author Response]

Essential revisions:1) (strong suggestion) The distinct modifications to BPTT/RTRL and the consequences of these modifications should be amplified in the existing text, to ensure that readers are sure to appreciate the results. For readers that are not familiar with gradient-based learning in RNNs and the ideas of random feedback, I fear that the separate ideas of gradient approximation and of random feedback might be muddled. Stressing that these two ideas are only related insofar as they separately address distinct shortcomings of existing algorithms would stress that they were introduced by the author not because they share a special relationship, but only to achieve that goal. This came across somewhat in the text, but I believe it could be amplified. Additionally, stressing earlier in the text why exactly finely-tuned feedback is a problem would be helpful (this might not be obvious to everyone). It's touched on in Discussion but explaining the concern in the Introduction would help frame that result more clearly.

To make the independence of the two approximations clear, the following sentence has been added to the Introduction: “While these two approximations address distinct shortcomings of gradient-based learning and can be made independently (as discussed below in Results), only when both are made together does a learning rule emerge that is fully biologically plausible in the sense of being causal, local, and avoiding fine tuning of feedback weights.”

To address the second point about the need for a clearer explanation of the problem with symmetric feedback weights, the following text has been added to the Introduction: “Such precise matching corresponds to fine tuning in the sense that it requires a highly particular initial configuration of the synaptic weights, typically with no justification as to how such a configuration might come about in a biologically plausible manner. Further, if the readout weights are modified during training of the RNN, then the feedback weights must also be updated to match them, and it is unclear how this might be done without requiring nonlocal information.”

2) (strong suggestion) Much of the math is relegated to the appendix. This seems like a pity since 1) the relationship of RFLO to BPTT is not obvious from the main text (undoubtedly requiring the reader to visit the appendix) and 2) the connection between RTRL and BPTT is presented so clearly (and, as the author noted, this connection has not been presented clearly in the literature). At a minimum, I would recommend migrating a minimal amount of the appendix to the main text to illustrate how RFLO relates to BPTT and RTRL. (Maybe, for example, Equation 9 could be presented in the main text). The tradeoff between BPTT and RTRL (trading-off non-locality and causality) and how RFLO addresses these non-biological features can only really be understood within the appendix, and I fear that not all readers will invest in it.

The author is thrilled by the reviewers’ suggestion to import more of the math from the Appendix into the main text. In response, two paragraphs following Equation 2 have been added, in which a minimal derivation of the RTRL learning rule is provided, and a precise discussion of the shortcomings of the learning rule and the two approximations made to ameliorate them is included. In addition, a more explicit encouragement for the reader to visit Appendix 1 for details about BPTT has been provided (“The other classic gradient-based algorithm, BPTT, involves a different approach for taking partial derivatives but is equivalent to RTRL; its derivation and relation to RTRL are also provided in Appendix 1.”) Details about BPTT and its relation to RTRL have been kept in the Appendix, however. This is because they are not essential for anything that follows in the main text, but rather are a bonus for the interested reader. Additionally, it would be impossible to explicate the topic clearly without introducing several more equations and a significant amount of technical discussion, providing a possible hurdle or annoyance to the less mathematically inclined of *eLife*’s readership.

3) Figure 2B seems to indicate that performance decreases consistently as the period grows longer and then perhaps asymptotes. It would be nice to know how performance for the RFLO changes as a function of the period, over a broader range of values (at least for this particular periodic task), to understand a reasonable timescale under which learning could occur. This is directly relevant to the author's proposal about how the brain might learn (i.e. through a sequence of actions) because it will dictate the duration of a "behavioral syllable" based on this learning rule.

This is an excellent observation. However, based on data that hasn’t been included in the manuscript, it appears that the existence of such a plateau is not a universal feature of the RNN performance as a function of task duration. Rather, it depends on quantities such as the number of training trials, network size, and the particular task that the RNN is being trained for. Taken together, there does not appear to be a universal timescale setting the duration of a behavioral syllable. Since it would take quite a lot of additional simulations to establish this definitively, and since the presumed result is a null one, the author’s preference would be not to pursue this point further. If the reviewers and editor feel strongly that this would be an important result, however, it could be done.

Other points:1) The theory showing that the learning rule decreases the error makes a number of assumptions: linear dynamics, uncorrelated neurons, and random weights. This seems very problematic: linear dynamics correlates the neurons, and the whole point of learning is to make the weights non-random (and presumably, correlated). Thus, it's not clear what the theory adds. At the very least, this should be pointed out in the main text. Even better (but not required) would be to try to verify, numerically, some of the expressions in the Appendix. That won't be particularly easy, but it seems possible. Alternatively, the change in error versus the alignment, and the alignment versus time, could be plotted during a simulation. This would go a long way toward supporting – or refuting – the theory.

The reviewer is certainly right that the limitations of the highly simplified theory in Appendix 2 should be highlighted more prominently in the main text. To address this, the following text has been added to the end of the first subsection in Results: “A number of simplifying assumptions have been made in the mathematical derivations of Appendix 2, including linear dynamics, uncorrelated neurons, and random synaptic weights, none of which will necessarily hold in a nonlinear network trained to perform a dynamical computation. Hence, although such mathematical arguments provide reason to hope that RFLO learning might be successful and insight into the mechanism by which learning occurs, it remains to be shown that RFLO learning can be used to successfully train a nonlinear RNN in practice.”

Regarding the reviewers’ other suggestion, the author has declined to attempt to verify numerically the expressions for the linearized network in the Appendix. On the one hand, it seems clear that strong quantitative agreement with the simulations of the nonlinear RNN would be too much to hope for, given the drastic simplifications and assumptions made in the derivation, as pointed out by the reviewer. On the other hand, it is equally clear that qualitative agreement between the theoretical expressions and simulations must occur, since the loss function does in fact decrease (Figure 2A) and the alignment between readout and feedback does in fact increase (Figure 2C) in simulations, as the theory predicts. Hence, it doesn’t appear that the numerical simulations would add much insight, and the reviewers’ leniency on this point is greatly appreciated

2) We couldn't find what the initial weights were in the learning rules. It would be good to know if small initial weights were needed, or if the learning rule works when the initial weights are large enough that the network is in the chaotic regime. (suggestion) In particular, a plot of performance versus initial weights (presumably the variance) would be informative.

A section has been added to the Materials and methods explaining weight initialization and other simulation details. Regarding the performance as a function of the initial weight variance, a supplementary figure to Figure 2 has been added, following the reviewers’ suggestion. In all other simulations, the initial weights were chosen to be sufficiently large to place the RNN in the chaotic regime. It is perhaps unsurprising that larger initial weights lead to better performance in a task such as the one shown in Figure 2, since only in this regime is the network able to autonomously generate rich time-varying signals.

3) In our experience, for local learning rules the variance in the error is large. The variance should be reported – not just the mean. In addition, more than 5 networks should be used to compute the error.

Following the reviewers’ suggestion, the number of networks used for the results in Figure 2 has been increased from 5 to 9.

Regarding the suggestion to report variance, the author’s opinion is that indicating the percentiles gives a clearer idea of the variability than standard deviation in logarithmic plots spanning many orders of magnitude, such as those shown in Figure 2. Hence, this format has been maintained in Figure 2 and the new supplemental figures related to it.

4) The networks were small (30 in Figure 2; 100 in Figure 3; not sure in Figure 4). What happens when the size increases? We're hesitant to ask for more simulations. However, if simulations with larger networks are not done, you need to be upfront about the fact that this study may not scale well to large networks.

As with other RNN training approaches, performance with RFLO learning generally *improves* for larger network sizes. Following the reviewers’ suggestion, a supplementary figure to Figure 2 has been added to show this. The reason that small networks have been used in simulations here is because such networks require less time to simulate, allowing for more training trials in a given amount of CPU time. A related point that was perhaps not sufficiently emphasized in the previous version of the manuscript is that the computational complexity of RFLO learning is on par with BPTT (both are ~N^2^ per timestep, as shown in Appendix 1), and greatly improved compared with RTRL (~N^4^ per timestep). Because this fact may be of practical importance for those who wish to implement the algorithm, it has been pointed out at the end of the subsection on Figure 2 (“As for other RNN training methods, performance of the trained RNN generally improves for larger network sizes […]”).

5) (Very important!) The Materials and methods section should contain all simulation details. As far as we can tell, some are missing: the initial values of the weights, the learning rates (after the grid search), explicit forms for the target functions in Figure 2, and the time step. And we may have missed other details; you should make sure that there's enough information that the simulations can be replicated. It's true that the code is supplied, but not all of us like to read code.

A section has been added to the Materials and methods explaining all simulation details.

6) You should also discuss Hoerzer et al. (2012, Cerebral cortex, 24(3), 677-690). This is an example of node perturbation without noise; it's instead based on overall performance relative to a running average. In that paper they train only the output weights, not the recurrent weights as is done here. However, if the network can do better than a training paradigm involving recurrent weights, it's worth mentioning. (suggestion) We would even go so far as to suggest comparing performance when training only the output weight against performance when training the recurrent weights.

The learning rule from the paper by Hoerzer appears to be a minor modification of that from Legenstein et al (2010), which was already discussed in the Discussion section. (Specifically, the learning rule in Legenstein is proportional to R-, where R is reward and is recent average reward, whereas Hoerzer uses just the sign of this value.) A citation to Hoerzer has been added to the updated manuscript, though there doesn’t seem to be a need for additional discussion beyond the what is already said about the Legenstein paper.

The suggestion to compare performance of an RNN in which only readout weights are trained with an RNN in which both recurrent and readout weights are trained is a good one. Following this suggestion, a supplementary figure has been added to Figure 2 showing that performance of an RNN in which both recurrent and readout weights are trained is better than that of an RNN in which only recurrent or only readout weights are trained. A paragraph discussing this has also been added to the main text (“It is also worthwhile to consider the relative contributions of the two types of learning in Figure 2, namely the learning of recurrent and of readout weights […]”)

7) The shaded regions in Figure 2 are only explained in the caption for panel D. It would be helpful to explain them in panel A as well.

The shaded regions have been explained in panel A.

8) The symbol 'i' is used to index over neurons in equation 1 (top), over outputs in the lower equation of equation 1 (and thus over the outputs in equation 2) and used to refer to the learning rates in the first sentence after equation 3. Given how technical the indexing can become, we would strongly suggest reserving 'i' (as you have for 'j','a', and 'b') for neurons only.

The indexing changes have been made following the reviewers’ suggestion.

9) The gray line in Figure 2B is lost in the text describing what each color represents. It should be enlarged and made more prominent.

The gray line has been made darker and thicker.

10) It would be helpful to add a more intuitive plotting convention for Figure 2C (such as a color gradient as τ gets longer).

The plot has been redrawn using a color gradient, as suggested by the reviewer.

11) (suggestion) In general, presenting the RFLO+Dale results in Figure 2 can be distracting from the main point (and the author doesn't treat it extensively in the text). We would suggest moving the Dale's Law results to a supplement to Figure 2 (see eLife's treatment of figure supplements).

The RFLO+Dale results have been moved to a supplemental figure, as suggested.

12) The last line of text before equation 5 seems to contain incorrect references to equations 14 and 15 of the appendix, which I believe are the same as equations 3 and 4 of the main text.

Thanks to the reviewer for pointing this out. The mistake has been corrected.

13) Figure 2D appears to have a color mismatch between the line and the explanatory text within the figure, for "Local only".

The color mismatch has been fixed.

14) The last sentence of "Interval matching" states that you will return to a point in the following subsection, but it's not clear which subsection you are referring to, or if that thread is ever discussed again later.

In order to make the logic clearer, the sentence has been replaced with the following: “In the following subsection, we shall address this shortcoming by constructing a network in which learned subsequence elements of short duration can be concatenated to form longer-duration sequences.”

15) The algorithm is only run on simple toy problems that are constructed for the manuscript. The experiments hint that RFLO struggles in the context of longer timescales, but the manuscript provides no grounding for how well the algorithm performs on richer data. It is important to see performance of the algorithm gauged on a commonly used problem from the machine learning literature. Various datasets could be used, but the language modeling task on the Penn TreeBank (PTB) is very well explored and serves as a kind of MNIST for sequence modeling. Here it is important to quantify success in a fashion that is congruent with current standards in ML.To be clear, high-level performance on such a task does not seem crucial. The paper is aimed at biology, and there is no reason to pursue top results. But it is important to be able to situate the obtained performance, and offer a benchmark for subsequent work on biologically plausible algorithms that simultaneously aim to be practical/functional.

Benchmarking the RFLO learning algorithm on a standard machine learning task such as PTB is an excellent idea. The main problem with this, though, is that training on such a task requires the use of a normalizing softmax on the RNN outputs together with a cross-entropy loss function (rather than mean squared error). In this case, additional nonlocal terms arise in the gradient-descent learning rule, forcing one to either (i) ignore these and try to use local RFLO anyway, or (ii) use a nonlocal version of RFLO learning that accounts for the different loss function and normalization. The first option leads (unsurprisingly) to terrible performance on PTB, while the second option is outside the scope of this study, since the point of the manuscript is the study of *local* RNN learning rules. Because of these considerations, results for the PTB task have regretfully not been added to the manuscript. Investigating whether local RNN learning rules for large-scale categorization tasks such as PTB could be an interesting question for future work, but it’s not currently obvious how this could be done.

Unfortunately, there don’t seem to be any similarly universal RNN benchmarking tasks of a sort that might provide a good test of RFLO learning. Presumably this is why other recent works on local RNN learning (e.g. Miconi, eLife 2017; Gilra and Gerstner, eLife 2018) haven’t applied their algorithms to a standard battery of tasks. Establishing such a battery would obviously be very useful for the field, but is unfortunately beyond the scope of the present work.

16) (weak suggestion, since this will be tough, and possibly beyond the scope of the work. but it would be nice if it could be done) Along these lines, and in the context of an externally defined problem, it would be ideal to see the performance of the algorithm explored using an LSTM architecture. Basic RNNs performance tends to be quite poor relative to LSTMs across many tasks. Is it easy to adapt the RFLO algorithm to more complex architectures, and does doing so deliver better performance on any task? The answer may simply be no; if so, that's OK.

The theory has so far not been applied to LSTMs due to concerns about the biological plausibility of LSTM architectures in the first place, regardless of the learning rule used. It is possible to derive local LSTM learning rules in a similar manner to RFLO learning, i.e. by dropping nonlocal terms that appear in the loss function gradient. The issue hasn’t been explored further in simulations, though, since the aim of the present paper is to develop biologically plausible learning rules for recurrent networks, not to study network architectures that have no clear basis in biology.

17) (weak suggestion) On this question of architecture: how well does RFLO function in the case that there are multiple 'layers' in the RNN (e.g. as in Deep LSTMs where the connectivity matrix is not all-to-all). Does the algorithm still function about as well, or does increasing the depth of the network slow training?

No significant benefit to using a two-layer architecture vs. a single layer with the same number of parameters was found for the task shown in Figure 2 (p=0.85 for n=9 networks, data not shown). It’s possible that multi-layer architectures could be more advantageous for more challenging tasks, for example tasks with compositional structure, such as the reach sequence task shown in Figure 4. Investigating whether this is the case, in addition to developing theoretical understanding of why multi-layer RNN architectures might be advantageous and investigating how they might be implemented in the brain (most obviously in pre- and primary motor cortex), is a fascinating direction for further study. Because these are big questions that would require an entire independent project to address in a satisfactory way, however, the author, with the editor’s permission, would prefer to defer this as future work.

18) More could be done to emphasize that the tasks solved go well beyond what is solvable using a feedforward network and feedback alignment. This is implicit in some of the tasks, but guiding the reader to see this clearly and conclusively would be ideal.

Following the reviewers’ suggestion, the following text has been added to the beginning of the Results section: “These tasks require an RNN to produce sequences of output values and/or delayed responses to an input to the RNN, and hence are beyond the capabilities of feedforward networks.”

19) In the original manuscript describing feedback alignment, convergence of error is proved under some very restrictive conditions. Please describe briefly, in the main text, how the theoretical results developed here are related to the original FA results. e.g. To what extend can they be seen as a generalization of those results? What assumptions are made differently or in addition to the FA results?

The mathematical results in Appendix 2 share some similarities with the FA results. Both approaches linearize the network, and the statistical average over RNN state vectors is similar to Lillicrap’s average over inputs. The result is not a straightforward extension of the Lillicrap result for a one-hidden-layer network, however, since the retaining of state information from one timestep to the next in our case makes it impossible to directly apply the feedforward results to an RNN “unrolled in time”. Specifically, the fact that the update to the recurrent weight matrix changes the RNN state vector trajectory makes the case considered here somewhat trickier. A few sentences about this have been added near the end of the first subsection in Results (“The mathematical approach for showing that alignment between readout and feedback weights occurs is similar to that used previously in the feedforward case […]”).

20) The manuscript briefly makes connections to the FORCE training method, but my feeling is that this currently doesn't go far enough. A few more sentences that summon more of the details of the model/algorithm from Sussillo and delineate the connections to RFLO would be useful to the reader.

A short paragraph on this topic has been added to the Discussion section (“In addition to the gradient-based approaches (RTRL and BPTT) already discussed above, another widely used algorithm for training RNNs is FORCE learning […]”.)

21) (suggestion) The section on 'Learning a sequence of actions' is interesting, but currently feels ad-hoc. The section almost feels more like a long discussion point than something that ought to sit in the results. The message is, I believe, that: RFLO and similar algorithms may suffer relative to ML approaches such as BPTT on longer time scales, but this is ok because there are ways to rescue performance for long sequences. In particular, winner-take-all and reward modulated Hebbian learning rules are introduced along with additional sets of neurons to rescue performance on a movement sequence task. There is a brief attempt to relate these to the literature on structures that connect to motor cortex, but this feels rushed. Ideally, the manuscript would develop this section further so that it can be appreciated both with respect to biology and ML. Additionally, on this note, it would be good to see the performance of BPTT on the full sequence problem without these ad-hoc approaches. There is certainly a limit to what BPTT can do: how much does is it struggle with this situation? Is the long sequence one the BPTT also struggles with? This would provide grounding for where RFLO and these additional ideas sit with respect to BPTT training in this more interesting case.

The reviewers’ suggestion to compare the performance of RFLO+subcortical loop with BPTT has been addressed with a new subpanel in Figure 4. This subpanel shows that, when the number of training trials is held constant, RFLO learning with the loop architecture outperforms not only RFLO learning without the loop architecture, but even outperforms BPTT.

22) There are a couple of citations that might be useful to include. For example, a mention of spiking variants of feedback alignment (e.g. "Deep learning with dynamic spiking neurons and fixed feedback weights"), and several recent works on approximations of RTRL in the ML literature that seem worth mentioning (e.g. "Unbiased online recurrent optimization", and "Approximating Real-Time Recurrent Learning with Random Kronecker Factors").

Thanks to the reviewers for pointing these references out. All three have been added to the first subsection of the Results section.

[Editors' note: further revisions were requested prior to acceptance, as described below.]Essential revisions:1) The treatment of Hoerzer et al. (a paper where you trained only the output weights) needs to be expanded. Our question last time was whether performance when only the output weights are trained could match performance when recurrent weights are trained. To address this, you trained only the output weights. However, as far as we could tell, there was no feedback. For a fair comparison, feedback weights are critical. Or at the very least, you should make sure you have a good weight initialization. (It is well known in the echo-state literature that when training only the output weights, the initialization of the other weight matrices is very important. The literature on echo-state networks contains lots of advice on how to ensure that these are set appropriately.) It would be nice if you included feedback weights and re-ran the simulations. That's not absolutely necessary, but if it's not done, then you will have to point out that you can't rule out the possibility that training the output weights actually works better than your approach of training the recurrent weights.

The simulation in which only the readout weights were trained did in fact include feedback of the readout, and the weight initialization is typical of what is used by Hoerzer et al. and in much of the echo state literature. This is already described in the Materials and methods section, but the reviewer is certainly correct that it should also be mentioned in the main text. A parenthetical note along these lines has therefore been added to the main text [“(with the readout fed back as an input to the RNN for stability – see Materials and methods)”].

2) It would be good to include your response to concern 15 (attempting RFLO learning on more complicated problems) in the Discussion of the submitted manuscript. Your response at present is satisfactory, but the insight you share into why extending RFLO learning to more complex problems is itself interesting, and will likely be interesting to readers of the paper.

The first paragraph of the Discussion section has been extended (“Another promising area for future work…”) to point out two points as promising avenues for future work: (i) stacked RNN architectures, and (ii) the topic that the reviewer suggested above, namely the possible application of local learning to discrete problems using cross-entropy loss functions and softmax normalization.